# Heme binding of transmembrane signaling proteins undergoing regulated intramembrane proteolysis

Thomas Kupke 1✉, Johann P. Klare 2 & Britta Brügger1✉

Transmembrane signaling proteins play a crucial role in the transduction of information across cell membranes. One function of regulated intramembrane proteolysis (RIP) is the release of signaling factors from transmembrane proteins. To study the role of transmembrane domains (TMDs) in modulating structure and activity of released signaling factors, we purified heterologously expressed human transmembrane proteins and their proteolytic processing products from *Escherichia coli*. Here we show that CD74 and TNFα are heme binding proteins. Heme coordination depends on both a cysteine residue proximal to the membrane and on the oligomerization of the TMD. Furthermore, we show that the various processing products have different modes of heme coordination. We suggest that RIP changes the mode of heme binding of these proteins and generates heme binding peptides with yet unexplored functions. The identification of a RIP modulated cofactor binding of transmembrane signaling proteins sheds new light on the regulation of cell signaling pathways.

[1] Heidelberg University Biochemistry Center, Heidelberg, Germany. [2] Department of Physics, University of Osnabrück, Osnabrück, Germany. ✉email: thomas. kupke@bzh.uni-heidelberg.de; britta.bruegger@bzh.uni-heidelberg.de

The sterol regulatory element-binding protein (SREBP) was the first described example of a membrane anchor acting to maintain transcription factors in a dormant state in the cytosol until their release is triggered by RIP[1]. RIP starts with cleaving off the extracellular domain of transmembrane proteins in a process called ectodomain shedding. Subsequently, the remaining membrane-bound fragment is processed by intramembrane-cleaving proteases such as signal peptide peptidase-like (SPPL)[2] proteases. Sequential proteolytic processing is now described for a wide range of integral membrane proteins including the single-spanning transmembrane proteins CD74 (also known as invariant chain) and tumor necrosis factor (TNFα). CD74 is involved in antigen presentation and macrophage migration inhibitory factor (MIF) signaling[3], and TNFα is a key regulator of the inflammatory response associated with many diseases including cancer, atherosclerosis, rheumatoid arthritis, and inflammatory bowel disease[4]. In case of CD74, cleavage in endosomal compartments generates the MHCII-associated CLIP peptide and a membrane-bound N-terminal fragment (NTF)[3]. Subsequent RIP of CD74-NTF by the SPPL protease SPPL2a generates the intracellular domain (ICD) involved in inward signaling[5]. In a similar proteolytic process, TNFα is cleaved by the sheddase TACE, liberating a soluble cytokine[6] and an NTF, which is further processed by SPPL2a/b to two different ICDs[7]. Further known substrates of human SPPL2a/b proteases are Fas ligand (FasL), Bri2, transferrin receptor-1 (Tfr1), Foamy virus envelope protein (FVenv, reviewed in ref. [8]), Neuregulin 1 (ref. [9]), CLN5 (ref. [10]), the lectin-like oxidized LDL receptor-1 (LOX-1)[11], and TMEM106B[12].

Although biological functions of CD74 and TNFα have been extensively studied, little is known about the role of their transmembrane domains in modulating structure and activity of the RIP-released fragments. In the present study we expressed in *Escherichia coli* full-length *CD74* and *TNFα* as well as proteolytic processing products of these transmembrane proteins. Unexpectedly, we observed that the type II transmembrane proteins CD74, TNFα, and other substrates of SPPL2a/b proteases such as Bri2, FasL, and Tfr1 are heme-binding proteins. Using biochemical and spectroscopic approaches, we elucidated the molecular mechanism of heme coordination of CD74 and TNFα. We describe that heme binding depends on both a cytosolic cysteine residue located proximally to the membrane and on the oligomerization of the TMD. Our data show that proteolytic processing of CD74 and TNFα changes the mode of heme binding, switching from a stable, hexa-coordinated ligation to a low-affinity, penta-coordinated heme bound by the short ICDs. We suggest that these short ICDs may function as intracellular heme sensor peptides participating in cellular signaling and speculate that full-length type II transmembrane proteins and their NTFs have so far unknown heme-dependent (enzymatic) functions.

## Results

### Single-spanning transmembrane protein CD74 is binding heme. To investigate the role of the transmembrane domain of SPPL2a/b substrates in RIP-mediated release of signaling and transcription factors[2], we first recombinantly expressed the TMD (aa 31–56) of human CD74 together with flanking residues including Cys28 in *E. coli* (CD74-TMD$_{(25–64)}$-10xHis). In full-length CD74 the TMD is preceded by a short cytoplasmic N-terminal domain, with the majority of the protein oriented on the luminal/extracellular side. During Ni-NTA chromatography CD74-TMD$_{(25–64)}$-10xHis elutes with an orange-brown color from the column (Fig. 1a, b). UV/VIS spectroscopy of purified CD74-TMD$_{(25–64)}$-10xHis (Fig. 1c) showed an absorption

spectrum with a Soret band at 415 nm (Fig. 1d) characteristic for heme-binding proteins. A Soret band around 415 nm is observed for proteins, which hexa-coordinate ferric ($Fe^{3+}$) heme via two histidine residues[13]. To exclude artificial heme binding via the His-tag fused to CD74-TMD$_{(25–64)}$, we replaced the 10xHis affinity tag in further experiments with maltose-binding protein (MBP).

MBP-tagged CD74 [MBP-CD74$_{(1–216)}$], containing the CLIP region and the soluble trimerization domain[3] (Fig. 1e, f), was expressed to test whether heme binding is also observed for the full-length protein. Solubilization of *E. coli* membranes and amylose chromatography was either performed in the presence of Triton X-100 or with n-dodecyl-β-D-maltopyranoside (DDM, not absorbing at 280 nm), since these non-ionic detergents do not disrupt the interaction between amylose and MBP (encoded by pMalC5X). In contrast to CD74-TMD$_{(25–64)}$-10xHis, the UV/VIS spectrum of MBP-CD74$_{(1–216)}$ showed a complex spectrum with absorbance maxima at 369, 422, and 444–446 nm (Fig. 1g), which is not typical for a heme-binding protein. However, in the presence of the His analog imidazole a split Soret/hyperporphyrin spectrum with maxima at 361 and 421 nm was obtained (Supplementary Fig. 1a), resembling the absorption characteristics of proteins with a heme bound via axial Cys and His residues[14].

### The NTF of CD74 binds heme via bis-thiolate ligation. During transport of MHC class II proteins to the plasma membrane CD74 is mainly processed by cathepsins to a membrane-bound NTF, CD74-NTF$_{(1–81)}$ (Fig. 1e)[3]. We therefore purified MBP-CD74-NTF$_{(1–81)}$ and observed a unique split Soret spectrum with absorbance maxima at 370, 448, and 553 nm and lacking the absorbance maximum at 422 nm seen for full-length CD74 (Fig. 1f, g). Notably, purified full-length MBP-CD74$_{(1–216)}$ contained significant amounts of a proteolytic degradation product that had approximately the size of CD74-NTF$_{(1–81)}$ fused to MBP (Fig. 1f; contributing to the observed UV/Vis spectrum shown in Fig. 1g), suggesting proteolytical processing by *E. coli* sheddases. So far only the DGCR8 (Pasha in insects and worms) heme protein has been described to show an analogous split Soret spectrum[15–17]. However, also ferric bis-thiolate hemin complexes[18] and sulfur donor complexes of thiolate-ligated heme enzymes (i.e. ferric enzymes with two thiolate groups as axial ligands)[19,20] show similar split Soret spectra. It was then shown that the DGCR8/Pasha dimer ligates heme via bis-thiolate coordination, with each monomer contributing one Cys residue for heme binding[15,17]. DGCR8/Pasha forms a complex with the RNase Drosha and the heme induced conformational change in DGCR8/Pasha enables high-fidelity microRNA processing by the Pasha-Drosha protein complex[16,21]. To examine whether heme is bound to oligomeric CD74-NTF as a bis-thiolate heme ligate as well, Cys28, the only Cys residue in MBP-CD74 fusions, was exchanged for Ala. Heme binding of MBP-CD74-NTF$_{(1–81)}$ C28A was drastically reduced and the split Soret spectrum characteristic for MBP-CD74-NTF$_{(1–81)}$ was no longer observed (Fig. 1f, g). The single point mutation C28A also drastically reduced the amount of heme bound to CD74-TMD$_{(25–64)}$-10xHis (Fig. 1a, c, d) and the Soret band was shifted from 415 to 412 nm. By reduction of ferric ($Fe^{3+}$) heme of MBP-CD74- NTF$_{(1–81)}$ to ferrous ($Fe^{2+}$) heme, the split Soret band was converted to a single Soret band at 426 nm, as was also observed for DGCR8, indicating that heme is now ligated by two non-deprotonated cysteine residues[17] (Fig. 1h). Gel filtration of full-length MBP-CD74$_{(1–216)}$ and MBP-CD74-NTF$_{(1–81)}$ confirmed that in both cases heme is stably bound (Supplementary Fig. 1b–f). Bis-thiolate ligation of heme by CD74-NTF was further confirmed by characterizing 10xHis-

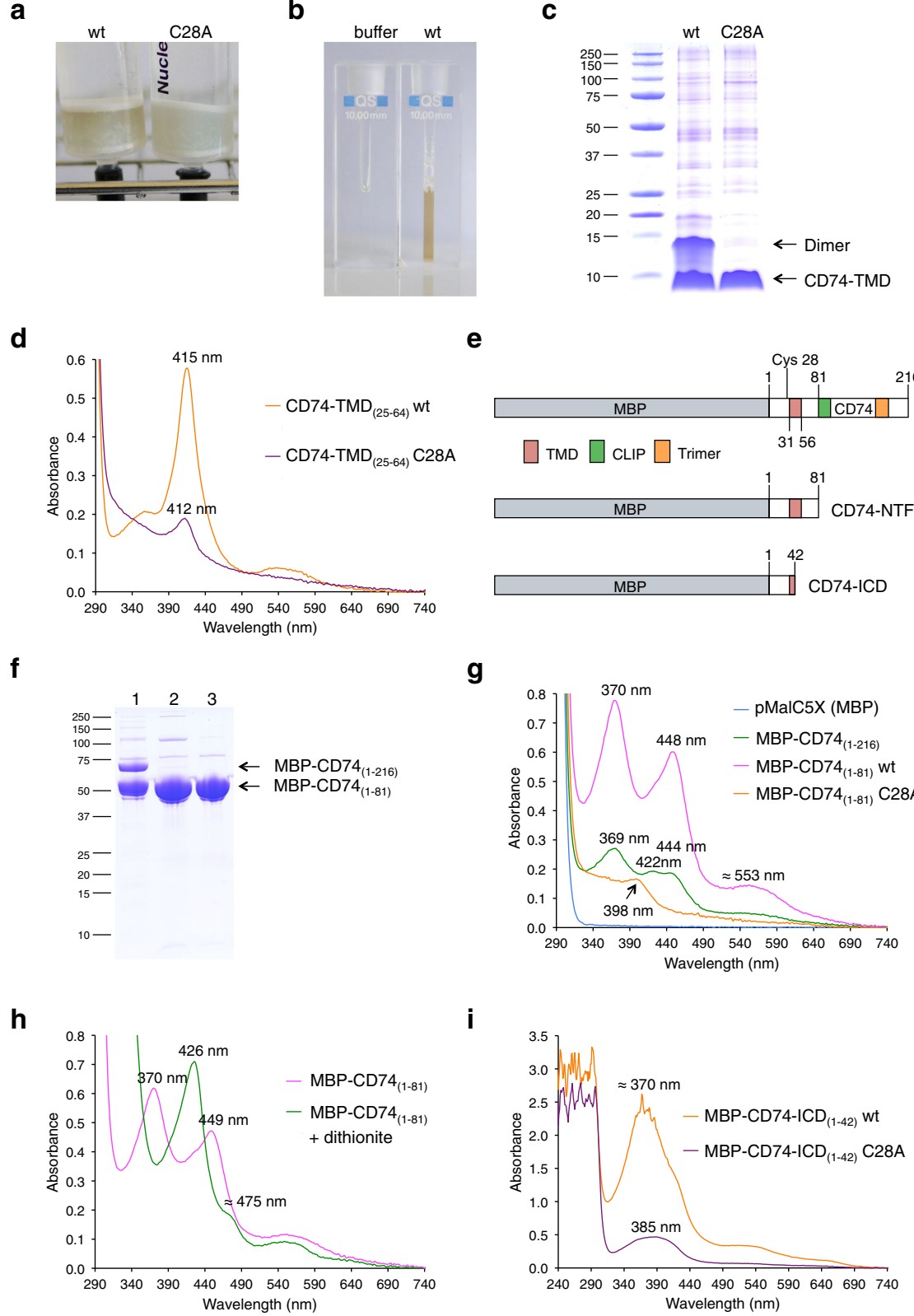

CD74-NTF. As observed for CD74-TMD-10xHis, 10xHis-CD74-NTF showed a single Soret band at 415 nm but decreasing the imidazole concentration changed the absorption spectrum to that of MBP-CD74-NTF (Supplementary Fig. 1g, h).

**CD74-ICD forms a penta-coordinate high-spin heme complex.** Cleavage of CD74-NTF by SPPL2a within the TMD generates

$ICD_{(1-42)}$[22–24] and removes the hydrophilic residues Gln48, Thr50, and Thr51, which contribute to the oligomerization of the TMD with interhelical hydrogen bonds[25]. Indeed, MBP-CD74-$ICD_{(1-42)}$ lacking these residues runs as a monomer in gel filtration [not shown, but compare results for MBP-TNFα-$ICD_{(1-34)}$]. As CD74 contains only a single cysteine residue, oligomerization of the TMD is a prerequisite for stable bis-

**Fig. 1 Heme binding and proteolytic processing of CD74. a** Peptides CD74-TMD$_{(25-64)}$-10xHis wt and TMD$_{(25-64)}$-10xHis C28A were purified by Ni-NTA chromatography. Ni-NTA columns are shown after the first washing step with buffer containing no imidazole. **b** Purified and concentrated CD74-TMD$_{(25-64)}$-10xHis wt has an orange-brown color. **c** Purification of CD74-TMD$_{(25-64)}$-10xHis proteins followed by SDS-PAGE and Coomassie staining. **d** UV/VIS spectroscopy of purified CD74-TMD$_{(25-64)}$-10xHis wt (orange trace) and CD74-TMD$_{(25-64)}$-10xHis C28A (violet trace) peptides. **e** MBP fusions used to characterize heme binding of CD74. The N-terminal TMD, the CLIP region, and the C-terminal trimerization domain are shown in different colors. **f** Purification of full-length MBP-CD74 and N-terminal fragments of CD74 was followed by SDS-PAGE and Coomassie staining: 1, MBP-CD74$_{(1-216)}$; 2, MBP-CD74-NTF$_{(1-81)}$ wt, and 3, MBP-CD74-NTF$_{(1-81)}$ C28A. **g** UV/VIS spectroscopy of MBP-CD74 fusion proteins purified from the membrane fraction using Triton X-100 as detergent. Blue trace, empty vector control; green trace, MBP-CD74$_{(1-216)}$; magenta trace, MBP-CD74-NTF$_{(1-81)}$ wt (purification with DDM as detergent led to the same UV/Vis spectrum); orange trace, MBP-CD74-NTF$_{(1-81)}$ C28A. **h** Dithionite reduction of bis-thiolate-ligated ferric heme to ferrous heme converted the split Soret band of MBP-CD74-NTF$_{(1-81)}$ to a single Soret band with an absorbance maximum at 426 nm. Magenta trace, MBP-CD74-NTF$_{(1-81)}$; green trace, MBP-CD74-NTF$_{(1-81)}$ plus dithionite. **i** Reconstitution of the intracellular domain with hemin. MBP-CD74-ICD$_{(1-42)}$ wt (orange trace) and MBP-CD74-ICD$_{(1-42)}$ C28A (violet trace) were reconstituted with hemin before amylose affinity chromatography. After elution from the amylose columns UV/VIS spectra of the reconstituted proteins were recorded (for more details see Supplementary Fig. 2). **a–d** Purification and UV/Vis spectroscopy of CD74-TMD-10xHis wt was performed three times, a representative experiment including CD74-TMD-10xHis C28A is shown. **f, g** MBP, MBP-CD74$_{(1-216)}$, MBP-CD74-NTF$_{(1-81)}$ wt, and C28A were purified in parallel and results for wt proteins were confirmed several times. **i** (Supplementary Fig. 2) Parallel heme reconstitution of MBP-CD74-ICD$_{(1-42)}$ wt and C28A proteins was done twice with the same outcome.

thiolate coordination of heme. MBP-CD74-ICD$_{(1-42)}$ should thus not exhibit the spectral features of bis-thiolate coordinated heme. Indeed, the heme cofactor does not copurify with MBP-CD74-ICD$_{(1-42)}$ (Supplementary Fig. 2a). However, MBP-CD74-ICD$_{(1-42)}$ could be reconstituted with hemin (i.e. ferric chloride heme) to form a penta-coordinate high-spin ferric heme complex with an axial cysteine thiolate as the fifth ligand, as shown by an absorbance maximum at 370 nm[14,26] and confirmed by the analysis of the C28A variant of MBP-CD74-ICD$_{(1-42)}$ (Fig. 1i, Supplementary Fig. 2a–h). In summary, our experiments show that CD74 is a heme-thiolate protein[27]. Binding of cysteine thiolate to ferric heme and low binding affinity of the cofactor localized at the protein surface are characteristics of heme sensor (i.e. heme responsive or heme sensing) proteins[14,28]. It is now well established that heme not only is the prosthetic group of oxygen transporting proteins hemoglobin and myoglobin, cytochrome P450 enzymes and the electron transport chain but has extended functions in gas sensing, sensing the cell redox state and by reversible binding to heme responsive proteins in transcriptional regulation[14]. The low heme-binding affinity is the reason that heme sensor proteins, in contrast to prototype heme proteins, are usually purified from cells without bound heme[14].

CD74 not only functions as an MHC class II chaperone[3] but has also been described as acting as a cell surface receptor for MIF[29]. After MIF-induced proteolytical processing of CD74 the ICD is translocated to the nucleus where it activates NFκB[30]. It is tempting to speculate that MIF-induced intramembrane proteolysis of CD74-NTF triggers the release of CD74-ICD to act as intracellular heme-sensing transcription factor.

**Transmembrane signaling protein TNFα is binding heme**. We then asked if heme binding to a type II transmembrane protein that undergoes SPPL2a/b-dependent proteolytical processing is a unique feature of CD74 or might also occur in other members of this family. Apart from TMEM106B all SPPL2a/b substrates have a cysteine residue either within the intracellular domain next to the TMD or within the N-terminal part of the TMD at the interface to the cytoplasm. As seen for CD74, we proposed that this cysteine residue enables stable heme binding of these proteins. We decided to first test human TNFα for heme binding. The TNFα-converting enzyme (TACE/Adam17) releases the soluble cytokine TNFα by cleaving the trimeric membrane-bound 233 aa precursor protein between residues 76 and 77 (Fig. 2a), generating the membrane-bound NTF$_{(1-76)}$ and soluble TNFα$_{(77-233)}$[7]. Secreted TNFα$_{(77-233)}$ forms trimers and binds to the membrane receptors TNFR1 and TNFR2 (ref. [7]). TNFα-NTF$_{(1-76)}$, with a conserved Cys residue in position 30 proximal

to the TMD (aa 31–53), can be further cleaved by the intramembrane protease SPPL2b between Cys49 and Leu50 (refs. [8,31]). Subsequent proteolysis by SPPL2a/b in a sequential manner releases two different ICDs, ICD$_{(1-34)}$ and ICD$_{(1-39)}$ (ref. [31]; Fig. 2a). The released ICDs then trigger the production of interleukin-12 by dendritic cells[32]. The various products of proteolytical processing of full-length TNFα were expressed as MBP fusions in *E. coli* and analyzed for heme binding (Figs. 2 and 3). Both full-length MBP-TNFα$_{(1-233)}$ and MBP-TNFα-NTF$_{(1-76)}$ showed split Soret spectra with maxima at 361 and 421 nm (Fig. 2b, c), suggesting that they are low-spin hexa-coordinated ferric heme-binding proteins with the heme ligand bound, using Cys as one the two axial ligands[14]. Thus, in contrast to CD74-NTF$_{(1-81)}$, heme binding of TNFα-NTF$_{(1-76)}$ did not occur by bis-thiolate ligation, as was also evident from the lacking absorbance maximum at 448 nm (Fig. 2c). Gel filtration of purified MBP-TNFα fusion proteins (Supplementary Fig. 3) showed that heme is stably bound by TNFα-NTF$_{(1-76)}$ (Supplementary Fig. 3a–c). To test whether heme binding in TNFα-NTF$_{(1-76)}$ indeed occurs via Cys and His residues as axial ligands of heme, we replaced the Cys residues C30 and C49 and the His residue H52 by alanine residues. Exchange of Cys30 of TNFα (corresponding to Cys28 of CD74) for Ala led to drastically reduced heme binding, whereas exchange of Cys49 within the TMD only led to a slight shift of the absorbance maximum to 421 nm (Fig. 2d–g). Exchange of His52 led to significant changes in the absorption spectrum compared to that of wt NTF (Fig. 2d–g). The heme content of TNFα-NTF$_{(1-76)}$ H52A was, however, not drastically reduced, suggesting that His52 is not an axial ligand of heme but is rather important for TMD structure and oligomerization (as supported by gel filtration analysis of MBP-TNFα-NTF$_{(1-76)}$ H52A; Supplementary Fig. 4). Therefore, the second axial ligand of heme in TNFα-NTF$_{(1-76)}$ is likely another, yet unidentified residue.

**Heme-promoted dimerization of TNFα-ICD$_{(1-39)}$**. We then investigated heme binding of TNFα-ICD$_{(1-39)}$. This ICD contains the N-terminal nine amino acid residues of the TMD, including an SXXS-based TMD-dimerization motif[33] which is absent in the shorter ICD$_{(1-34)}$ (Fig. 3a). In contrast to MBP-TNFα-NTF$_{(1-76)}$, we observed for MBP-TNFα-ICD$_{(1-39)}$ absorbance maxima at 371, 451, and 553 nm characteristic for ferric heme binding via bis-thiolate coordination (Fig. 3b–f, Supplementary Fig. 3d–g). This result indicated that MBP-TNFα-ICD$_{(1-39)}$ occurred as dimer and that removal of C-terminal residues of the NTF led to intermolecular heme ligation involving Cys30. Exchange of Cys30 for Ala thus removes both axial ligands of the heme cofactor. Consequently, the heme content of MBP-TNFα-ICD$_{(1-39)}$ C30A

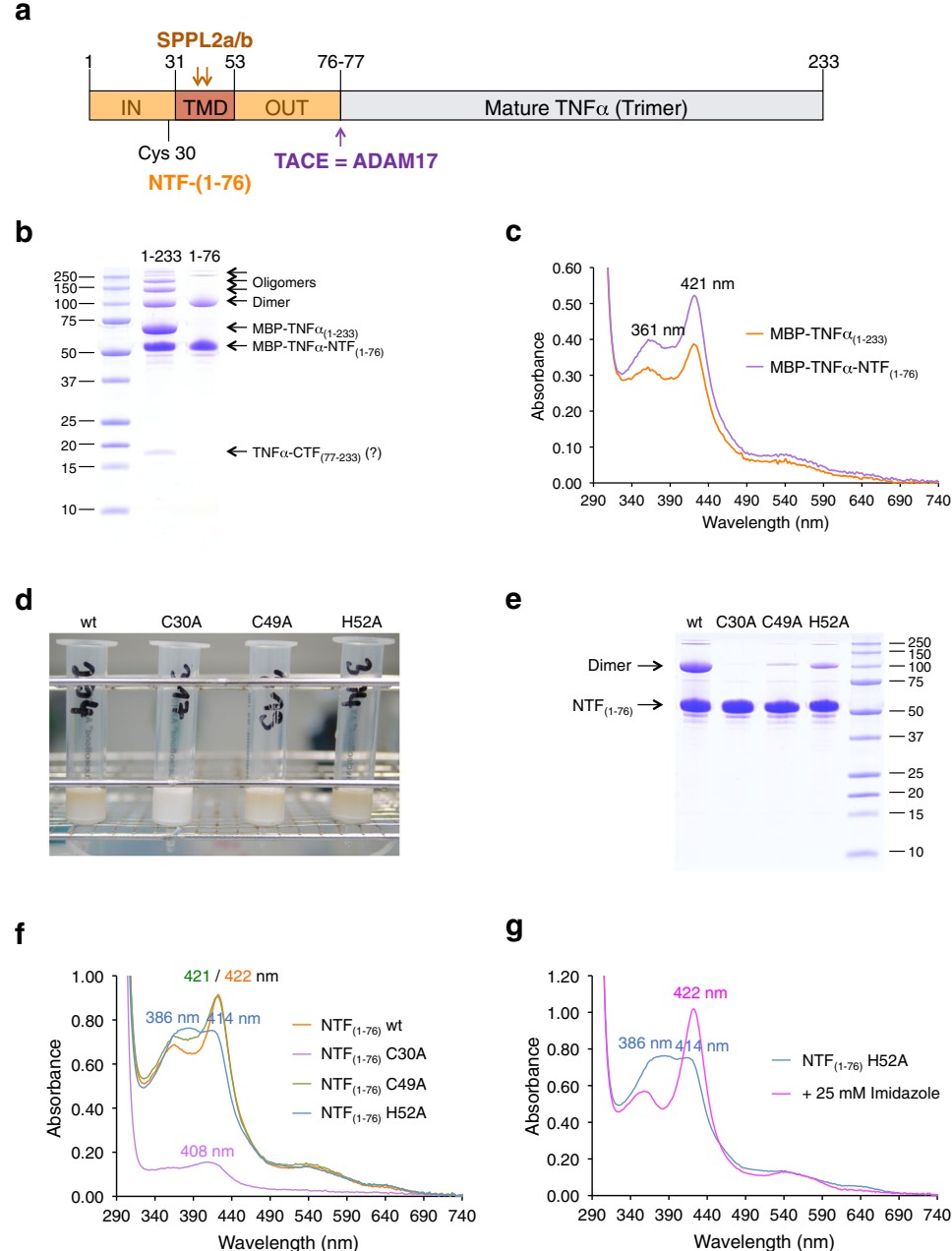

**Fig. 2 TNFα is a heme-binding protein. a** Domain structure and proteolytical processing of TNFα [7]. **b** Purification of full-length MBP-TNFα(1–233) and MBP-TNFα-NTF(1–76) was followed by SDS-PAGE. MBP-TNFα(1–233) was partially degraded to a protein with the same size as MBP-TNFα-NTF(1–76) and a (copurified) 17 kDa protein (present as double band) which was most likely mature TNFα. **c** UV/VIS spectroscopy of purified MBP-TNFα(1–233) (orange trace) and MBP-TNFα(1–76) (violet trace). **d** Amylose chromatography of wt and mutant (C30A, C49A and H52A, respectively) MBP-TNFα-NTF(1–76). Amylose-affinity columns are shown after loading the columns with Triton X-100 solubilized membrane proteins and subsequent washing. **e** Purification of wt and mutant MBP-TNFα-NTF(1–76) proteins was followed by SDS-PAGE. **f** UV/VIS spectroscopy of wt and mutant (C30A, C49A, and H52A, respectively) MBP-TNFα-NTF(1–76) proteins; absorbance maxima are indicated. **g** To confirm that also MBP-TNFα-NTF(1–76) H52A bound heme, 25 mM imidazole was added to generate a Soret band at 422 nm. All proteins (apart from MBP-TNFα-NTF(1–76) C30A, which was purified only once) shown were purified at least twice and representative UV/Vis spectra are shown. Gel filtration of MBP-TNFα-NTF(1–76) wt and mutant H52A protein was done twice with the same outcome (see Supplementary Fig. 4).

was very low (Fig. 3b–e). The heme content of MBP-TNFα-ICD(1–39) (and of all the other proteins investigated in this study) could be increased by supplementing the growth medium of *E. coli* with the heme precursor δ-aminolevulinic acid (Supplementary Fig. 5a–d). The rate of protein synthesis is higher than the rate of heme biosynthesis, so that expression in *E. coli* in general does not lead to pure holoproteins. However, copurification of heme with TNFα and CD74 indicates high-affinity

binding, since the labile heme concentration in cells is in the nanomolar range[34].

Gel filtration of MBP-TNFα-ICD(1–39) revealed that the ICD occurred in three different oligomeric states (Supplementary Fig. 3d–g): as a high-molecular weight cluster, as a dimer binding heme via bis-thiolate coordination, and as monomer that does not stably bind heme. The dimer content of TNFα-ICD(1–39) wt was increased when the growth medium had been supplemented

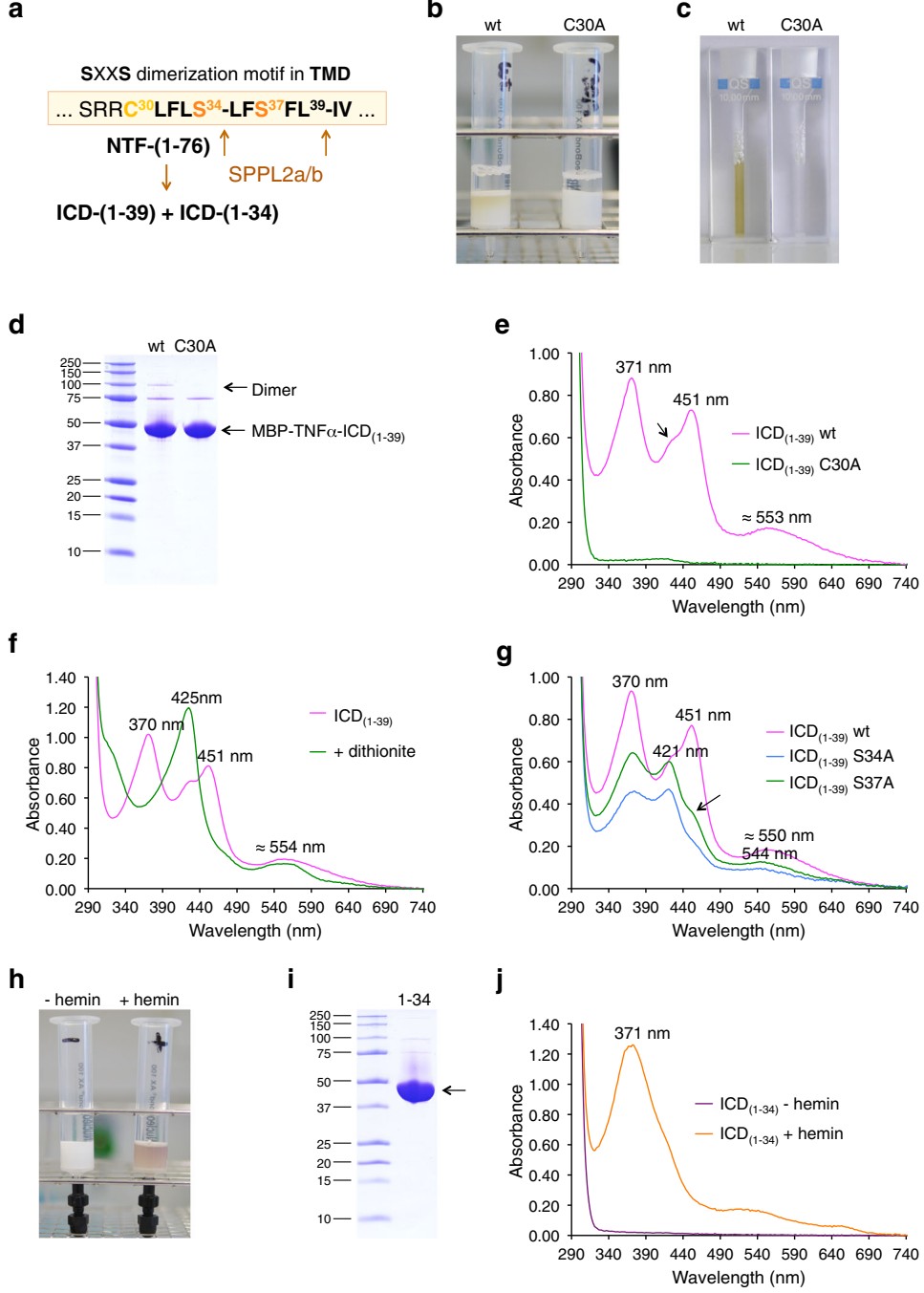

**Fig. 3 Heme binding of ICDs generated by intramembrane cleavage of TNFα-NTF by SPPL2a/b. a** Cleavage sites of SPPL2a/b within the TMD of TNFα-NTF$_{(1-76)}$ and the putative SXXS TMD-dimerization motif. **b** Purification of MBP-TNFα-ICD$_{(1-39)}$ proteins (wt and C30A mutant) by amylose chromatography (all proteins shown in Fig. 3b-g were purified from the membrane fraction using Triton X-100). **c** Purified wt but not mutant C30A MBP-TNFα-ICD$_{(1-39)}$ is a yellow-green protein. **d** Purification of wt and mutant C30A MBP-TNFα-ICD$_{(1-39)}$ proteins was followed by SDS-PAGE. **e** UV/VIS spectroscopy of purified wt (magenta trace) and mutant C30A (green) MBP-TNFα-ICD$_{(1-39)}$ proteins. The split Soret band with absorbance maxima at 371 and 451 nm indicates bis-thiolate ligation of ferric heme. The arrow indicates a shoulder at about 420–425 nm in the absorption spectrum of ICD$_{(1-39)}$, which results from high-molecular weight protein clusters (see Supplementary Fig. 3d). **f** Dithionite reduction of ferric heme (magenta trace) to ferrous heme (green trace) converted the split Soret band of MBP-TNFα-ICD$_{(1-39)}$ to a single Soret band with an absorbance maximum at 425 nm. **g** UV/VIS spectroscopy of purified MBP-TNFα-ICD$_{(1-39)}$ wt (magenta trace) and mutants S34A (blue trace) and S37A (green trace). **h** Purification and reconstitution of MBP-TNFα-ICD$_{(1-34)}$. Cytosolic protein solutions of *malE-TNFα-ICD-(1-34)*-expressing *E. coli* cells were either directly applied to amylose affinity chromatography (−*hemin*) or reconstituted with 50 μM hemin (+*hemin*) before amylose affinity chromatography. **i** Purified MBP-TNFα−ICD$_{(1-34)}$ was analyzed by SDS-PAGE. **j** UV/VIS spectra of MBP-TNFα-ICD$_{(1-34)}$ (violet trace) and hemin-reconstituted MBP-TNFα-ICD$_{(1-34)}$ (orange trace). MBP-TNFα-ICD$_{(1-39)}$ wt was purified at least ten times (using different conditions: for purification from the membrane fraction using Triton X-100 and DDM as detergents, from the cytoplasmic fraction without detergent). In all cases split Soret spectra were observed, figures **e–g** show UV/Vis spectra of MBP-TNFα-ICD$_{(1-39)}$ wt from three different purifications. Purification and characterization of mutant MBP-TNFα-ICD$_{(1-39)}$ S34A and S37A proteins were repeated (including the detailed analysis shown in Supplementary Fig. 6) and representative results are shown. **h–j** Hemin reconstitution of MBP-TNFα-ICD$_{(1-34)}$ was done once.

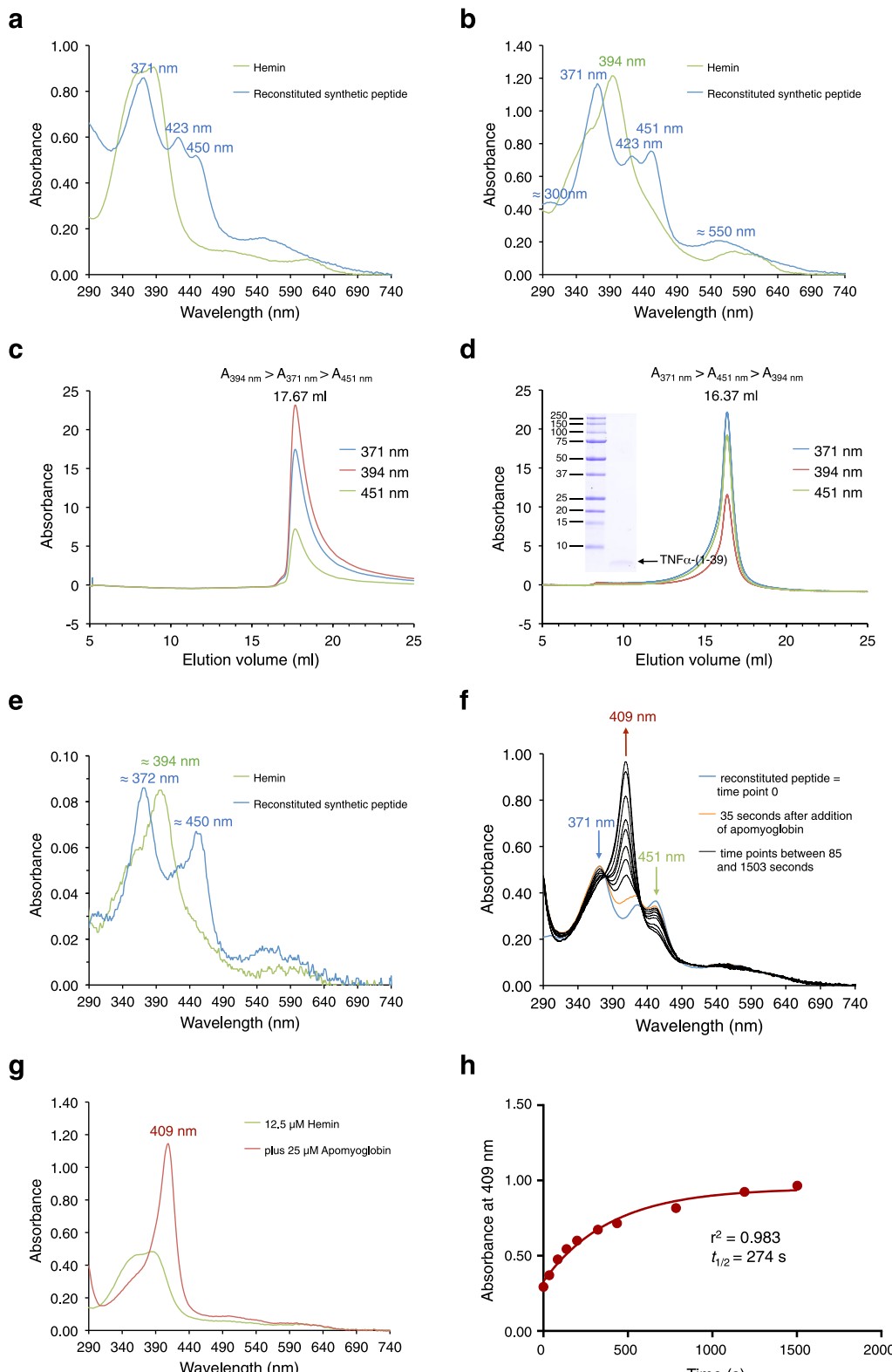

with δ-aminolevulinic acid (Supplementary Fig. 5e), whereas the non-heme-binding mutant protein TNFα-ICD$_{(1–39)}$ C30A was mainly present as monomer (Supplementary Fig. 5f, g). We concluded that heme binding promotes dimerization of TNFα-ICD$_{(1–39)}$. When MBP-TNFα-ICD$_{(1–39)}$ was purified from the cytoplasmic fraction without detergent we observed absorbance maxima at 370, 426, and 451 nm (Supplementary Fig. 5h, i).

To investigate the importance of the SXXS motif for heme-promoted dimerization, we analyzed the MBP-TNFα-ICD$_{(1–39)}$ mutants S34A and S37A (Fig. 3g, Supplementary Fig. 6). Exchange of either Ser34 or Ser37 for Ala led to significantly decreased absorbance at 451 nm and the appearance of an absorption band at 421 nm, indicating that less bis-thiolate coordinated heme and thus less dimer were present (Fig. 3g).

**Fig. 4 Hemin reconstitution of synthetic TNFα-(1–39) peptide and heme dissociation by addition of apomyoglobin. a** UV/VIS spectra of 25 μM hemin (green trace) and hemin-reconstituted synthetic TNFα-(1–39) peptide (blue trace) in TN buffer. **b** UV/VIS spectra of 25 μM hemin (green trace) and hemin-reconstituted synthetic TNFα-(1–39) peptide (blue trace) in TN buffer containing 0.1% Triton X-100. **c** Gel filtration of hemin (in TN/0.1% Triton X-100) and **d** of hemin-reconstituted synthetic TNFα-(1–39) peptide (in TN/0.1% Triton X-100) using a Superose 6 Increase column equilibrated in TN buffer containing 0.1% Triton X-100. Elution was followed by absorbance at 371 nm (blue trace), 394 nm (red trace), and 451 nm (green trace). Binding to the synthetic peptide shifts the elution volume of heme from 17.67 to 16.37 ml as confirmed by analysis of the peak fraction by SDS-PAGE (inset in **d**). **e** Peak fractions of both gel filtrations were analyzed by UV/Vis spectroscopy (hemin, green trace; hemin-reconstituted peptide, blue trace) confirming stable bis-thiolate ligation of heme by the synthetic peptide. **f** Hemin-reconstituted synthetic TNFα-(1–39) peptide (containing about 12.5 μM bis-thiolate-ligated heme) (blue trace) was incubated at room temperature with about 25 μM apomyoglobin in TN buffer pH 8.8 and UV/VIS spectra were recorded for the next 25 minutes (orange trace, after 35 s; black traces, time points between 85 and 1503 s). Formation of metmyoglobin led to a decrease of absorbances at 371 (blue arrow) and 451 nm (green arrow) and an increase of absorbance at 409 nm (red arrow). **g** In a control experiment incubation of 12.5 μM hemin (green trace) with about 25 μM apomyoglobin in TN buffer pH 8.8 immediately led to formation of metmyoglobin with an absorbance of 1.15 at 409 nm (red trace). **h** The 409 nm (red trace) absorbances of experiment **f** were plotted against the incubation time. The data were then fitted with the GraphPad Prism program using the exponential one phase decay function giving a decay half-life of $t_{1/2} = 274$ s. Three independent reconstitution experiments were performed (**a**, **b** and **f**). **f–h** show data from one experiment. In a second heme dissociation experiment $t_{1/2}$ was determined to be 290 s.

Separation of the different oligomeric forms of ICD$_{(1–39)}$ by gel filtration confirmed that for the wt protein the dimer to cluster ratio is more than twice that of the mutants (Supplementary Fig. 6c–g). In summary, heme-promoted dimerization of MBP-TNFα-ICD$_{(1–39)}$ not only requires a cysteine residue per monomer contributing thiolate as an axial heme ligand but also the SXXS dimerization motif S$^{34}$LFS$^{37}$FL.

**Deletion of TMD residues changes heme-binding properties.** MBP-TNFα-ICD$_{(1–34)}$ in which Ser37 and most of the remaining TMD are deleted was purified without bound heme (Fig. 3h–j). However, as observed for MBP-CD74-ICD$_{(1–42)}$, MBP-TNFα-ICD$_{(1–34)}$ could be reconstituted with hemin to show a single Soret band at 371 nm (Fig. 3j). Thus, in contrast to TNFα-ICD$_{(1–39)}$ heme is bound in this shorter ICD only by one cysteine residue. We assume that not only Cys (as axial ligand) but also hydrophobic residues originated from the TMD contribute to heme binding of the ICDs by contacting hydrophobic side chains of heme. Hemin-reconstituted MBP-TNFα-ICD$_{(1–34)}$ lost the cofactor during gel filtration and was present as monomer (Supplementary Fig. 3h, i), confirming that residues 35–39 (and Ser34) are important for dimerization. Together these data suggest that TNFα-ICD$_{(1–34)}$, like TNFα-ICD$_{(1–39)}$, is a heme sensor peptide but with different heme associating and dissociating properties.

**Reconstitution of bis-thiolate heme ligation.** To confirm that bis-thiolate heme ligation is an intrinsic property of TNFα-ICD$_{(1–39)}$, we reconstituted synthetic peptide TNFα-(1–39) with hemin and subsequently purified the peptide–heme–peptide complex by gel filtration (Fig. 4a–e). Addition of apomyoglobin to the hemin-reconstituted peptide then led to dissociation of heme with a half-life of about 4.6 min (Fig. 4f–h), whereas heme cofactor of DGCR8 could not be dissociated by incubation with apomyoglobin[15]. Reversible association with heme and heme-promoted dimerization support the idea that TNFα-ICD$_{(1–39)}$ is a heme sensing and heme responsive signaling peptide. To our knowledge, TNFα-(1–39) is the first described sensor that ligates heme by bis-thiolate coordination.

**Stabilization of bis-thiolate ligation of heme by a CP motif.** In many heme sensor proteins the heme-binding cysteine residue is followed by a proline residue, which is thought to increase the binding affinity of the cysteine residue for ferric heme[28,35,36]. To increase the heme content of TNFα-ICD$_{(1–39)}$ for structural studies of bis-thiolate-ligated heme proteins, we introduced this CP motif into TNFα-ICD$_{(1–39)}$ by exchanging Leu31 for Pro.

Compared to purified wt MBP-TNFα-ICD$_{(1–39)}$ protein the L31P mutant has a roughly twofold higher heme content (Fig. 5a–c) without changing the heme coordination site (in both cases split Soret spectra with maxima at 371 and 451 nm were observed). To verify that introduction of a proline residue increases the affinity of Cys for ferric heme, wt and L31P proteins were titrated with increasing imidazole concentrations. The histidine analog imidazole replaces one cysteine ligand of the heme cofactor converting the split Soret spectrum of bis-thiolate-ligated heme to a Soret band at 425 nm (Fig. 5d, e; replacement of both cysteine residues at higher imidazole concentrations would lead to a Soret band at about 415 nm; compare Supplementary Fig. 1h). It was also observed for ferric bis-thiolate hemin complexes that imidazole, even at 10-fold excess, exchanges only one thiolate[18]. Disruption of bis-thiolate heme ligation of wt protein occurs at lower imidazole concentrations than observed for the mutant L31P protein. This higher affinity of imidazole for heme bound by wt protein (equals weaker interaction of Cys with heme) is reflected by a $K_d$ value of 2.4 mM compared to 10.0 mM for mutant L31P protein (Fig. 5f). This shows that the L31P mutation stabilizes the interaction of cysteine with heme and by this bis-thiolate heme ligation. The heme content of MBP-TNFα-ICD$_{(1–39)}$ was then further increased by decreasing the growth temperature of the used *E. coli* cells from 30 to 20 °C (to slow down rate of protein synthesis; Fig. 5g) leading to $A_{451}/A_{280}$ ratios of about 0.25 instead of 0.18 (compare Fig. 5b). MBP-TNFα-ICD$_{(1–39)}$ L31P purified under these conditions was concentrated and used for electron paramagnetic resonance spectroscopy (EPR) at 10 K (Fig. 5h). MBP-TNFα-ICD$_{(1–39)}$ L31P showed a rhombic, low-spin ($S = ½$) signal centered at $g \approx 2$ with narrow separation between $g_x$, $g_y$, and $g_z$ which is typical for hexa-coordinated heme thiolates[27]. The observed g values of $g_z = 2.40$, $g_y = 2.28$, and $g_x = 1.91$ are within the range of g values not only of ferric bis-thiolate hemin complexes[18] but also of ferric low-spin sulfur donor complexes of the thiolate-ligated heme enzymes cytochrome P450 ($g_z = 2.36–2.50$, $g_y = 2.24–2.27$, $g_x = 1.89–1.95$) and chloroperoxidase ($g_z = 2.37–2.45$, $g_y = 2.25–2.27$, $g_x = 1.81–1.91$) (reviewed in ref. [37]) further supporting bis-thiolate ligation of MBP-TNFα-ICD$_{(1–39)}$.

**The FCLLH motif is important for oligomerization of TNFα-NTF.** We then asked if N-terminal TNFα-peptides having more than nine TMD residues (Supplementary Fig. 7a) also coordinate heme by bis-thiolate ligation and which residues of the NTF are important for trimerization and thus for heme ligand switching. Therefore, we analyzed heme binding and oligomerization of NTFs with increasing C-terminal deletions by UV/VIS spectroscopy and gel filtration, respectively (Supplementary Fig. 7b–d).

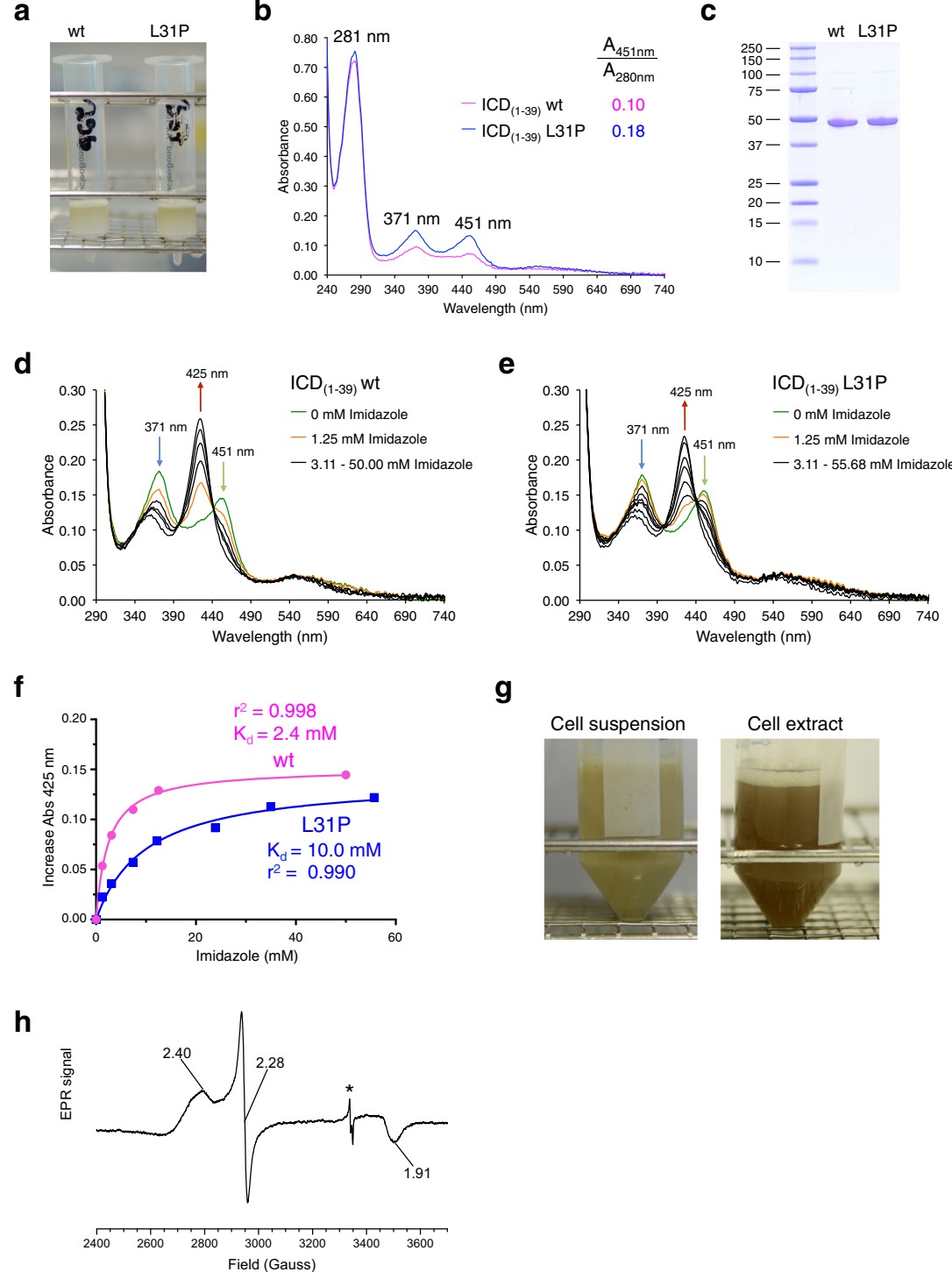

The UV/VIS spectrum of TNFα-NTF$_{(1-54)}$ was very similar to that of TNFα-NTF$_{(1-76)}$ wt; however, additional deletion of His$^{52}$-Phe$^{53}$-Gly$^{54}$ changed the UV/VIS spectrum of TNFα-NTF$_{(1-51)}$ to that of TNFα-NTF$_{(1-76)}$ H52A. In contrast to TNFα-NTF$_{(1-51)}$ which eluted in gel filtration experiments similar to TNFα-NTF$_{(1-76)}$, TNFα-NTF$_{(1-47)}$ formed no trimers and eluted as clusters and as dimers. This indicates that the motif Phe$^{48}$-Cys$^{49}$-Leu$^{50}$-Leu$^{51}$-His$^{52}$ including the primary SPPL2b cleavage site Cys$^{49}$-Leu$^{50}$ (refs. [8,31]) is important for trimerization and proper heme binding by TNFα-NTF$_{(1-76)}$. From all the investigated constructs only TNFα-ICD$_{(1-39)}$ predominantly formed dimers coordinating heme by bis-thiolate ligation.

**A new family of heme-binding transmembrane proteins.** In a final set of experiments we tested whether heme binding extends to other SPPL2a/b substrates with TMD-proximal Cys residues. We analyzed the anti-Alzheimer protein Bri2 (Fig. 6a, b), the FasL protein (Fig. 6a, b), lectin-like oxidized LDL receptor-1 LOX-1 (Fig. 6c, d), and the transferrin receptor Tfr1 (Fig. 6e–i). All these proteins have essential functions in cells. The Bri2 protein is associated with familial British and Danish dementias and reduces fibrillization of the amyloid-β-peptide[38]. The dimeric transferrin receptor imports iron into cells by binding holo-transferrin[39] and binding of FasL to the Fas receptor FasR leads to apoptosis[40]. As assumed these proteins are indeed heme-binding

**Fig. 5 The L31P mutation stabilizes bis-thiolate ligation of heme by TNFα-ICD$_{(1-39)}$. a** Purification of wt and L31P MBP-TNFα-ICD$_{(1-39)}$ proteins by amylose chromatography using DDM as detergent. **b** UV/VIS spectroscopy of purified (not concentrated) wt (magenta trace) and mutant L31P (blue) MBP-TNFα-ICD$_{(1-39)}$ proteins. The ratio of absorbance at 451 nm (bis-thiolate-ligated heme) and 280 nm (protein and heme) shows the higher heme content of the L31P mutant (0.18 vs 0.10 for wt protein). **c** Purification of wt and mutant L31P MBP-TNFα-ICD$_{(1-39)}$ proteins was followed by SDS-PAGE. **d** wt and **e** mutant L31P MBP-TNFα-ICD$_{(1-39)}$ proteins were purified using DDM as detergent, adjusted by dilution of the L31P mutant to the same heme concentration and then incubated with increasing imidazole concentrations up to about 50 mM. Decrease of absorbances at 371 nm (blue arrow) and 451 nm (green arrow) and simultaneous formation of a Soret band at 425 nm (red arrow) was observed. **f** Absorbance at 425 nm of the experiments shown in **d** and **e** was plotted against the imidazole concentration to analyze the strength of interaction between cysteine and heme of wt (magenta trace) and mutant L31P (blue trace) proteins. The data were then fitted using the saturation binding equation One site specific binding of the GraphPad Prism program and $K_d$ values for wt (magenta) and L31P mutant (blue) were determined. **g** Heme incorporation into MBP-TNFα-ICD$_{(1-39)}$-L31P was optimized by overnight expression at 20 °C. After sonification of the yellow-green *E. coli* cells (left) a brown-green extract (right) was obtained. **h** X-Band continuous wave EPR spectrum of MBP-TNFα-ICD$_{(1-39)}$-L31P recorded at 10 K with *g* values marked. An artifact originating from the EPR resonator at 3340 Gauss is labeled by an asterisk. The higher heme content of L31P mutant compared to wt MBP-TNFα-ICD$_{(1-39)}$ was confirmed in several experiments; a representative experiment is shown. **d–f** MBP-TNFα-ICD$_{(1-39)}$ wt and L31P were purified in parallel and imidazole titration was done under comparable conditions. **h** In a second EPR experiment *g* values of 2.41, 2.28, and 1.92 were obtained for MBP-TNFα-ICD$_{(1-39)}$ L31P.

proteins (Fig. 6b, d, g, h) with Tfr1-NTF$_{(1-100)}$ binding heme via bis-thiolate coordination as indicated by the UV/Vis spectrum with a split Soret band displaying absorbance maxima at 372 and 453 nm (Fig. 6g). However, the SPPL2a/b substrate LOX-1 containing a cysteine residue within the N-terminal part of the TMD and not within the intracellular domain next to the TMD is binding heme only to a very low extent compared to Bri2-NTF (Fig. 6d). Interestingly, exchange of each of the two cytosolic cysteine residues of Tfr1 led to disruption of bis-thiolate ligation of heme (Fig. 6g, h). We conclude that in contrast to CD74 and TNFα two different cysteine residues, namely Cys62 and Cys67, are axial heme ligands. Heme bis-thiolate coordination of MBP-Tfr1-NTF$_{(1-100)}$ in contrast to that of MBP-CD74-NTF$_{(1-81)}$ and MBP-TNFα-ICD$_{(1-39)}$ is mostly disrupted by using DDM instead of Triton X-100 as detergent for purification (Fig. 6g, violet trace). It is conceivable that DDM impairs dimerization of Tfr1-NTF and that Cys62 from one monomer and Cys67 from the other monomer are axial heme ligands. However, further research is necessary to elucidate if heme ligation occurs within one monomer or between two monomers of the Tfr1 dimer.

We propose that as described above for CD74 and TNFα RIP will change the heme binding mode of Bri2, FasL, and Tfr1 (with additional cytosolic proteolytic processing between Cys62 and Cys67?) and generates heme sensor peptides.

We suggest to classify CD74, TNFα, and Tfr1 (and Bri2, FasL, and other SPPL2/b substrates as additional putative members) as a new family of heme-binding proteins, called THOMAS proteins (**T**hiolate-**H**eme **O**ligomeric type II trans-**M**embrane proteins with heme binding mode **A**djusted by **S**PPL2a/b, Fig. 7). We assume that proteolytic processing products binding heme via bis-thiolate ligation (Fig. 7a) using cytosolic Cys residues located next to the TMD as axial ligands (Fig. 7b) are characteristic for THOMAS proteins (Fig. 7c).

## Discussion

Heme concentration in cells is decreased by degradation of heme by HO-1 to biliverdin, carbon monoxide, and iron. HO-1 is an endoplasmic reticulum anchored enzyme facing the cytosol[41]. Recently, it was reported that HO-1 mitigates pulmonary tuberculosis caused by *Mycobacterium tuberculosi*[42] and in a second study that patients with SPPL2a deficiency have an increased susceptibility to mycobacterial infections due to toxicity of the accumulated CD74-NTF[43]. This toxicity might be related to a yet undefined biological function of CD74-NTF-heme. HO-1 also attenuates TNFα-dependent cell injury and apoptosis[44]. We speculate that HO-1-dependent heme degradation leads to reduced binding of heme to TNFα and its proteolytic processing products, thereby reducing the level of heme sensor/effector

peptides. On the other hand, it is suggested that increased expression of HO-1 plays a role in progression of neurodegeneration and carcinogenesis[45]. Further research will help to clarify, whether some of the contradicting effects of HO-1 on the progression of diseases are indeed related to heme binding of transmembrane signaling proteins such as CD74, TNFα, the anti-Alzheimer protein Bri2, Fas ligand, and Neuregulin 1.

Our data suggest that the mode of heme binding by cysteine residues changes during regulated intramembrane processing of the trimeric THOMAS proteins CD74 and TNFα (Fig. 8a) and during their assembly (Fig. 8b). Cysteine residues not used for heme binding would be available for competing reversible post-translational palmitoylation (Cys28 in CD74 and Cys30 in TNFα)[3,46] adding another level of regulation. During regulated intramembrane processing these type II transmembrane proteins are either cleaved by soluble proteases (CD74) or by plasma membrane located sheddases (TNFα). Removal of the trimeric ecto-domain (or in case of TNFα of the ectodomain and C-terminal residues of the NTF) leads to a heme ligand switch within the NTF, resulting in stable heme ligation via bis-thiolate coordination. In CD74-NTF oligomers (dimers or trimers[25]) two (of the three) cysteine residues contact ferric heme. Further processing of the NTF leads to formation of intracellular soluble ICDs with heme sensor properties. Proteolytical processing of TNFα results in two ICDs, ICD$_{(1-39)}$ and ICD$_{(1-34)}$. These two ICDs differ in their mode of heme binding and thus might also evoke different signaling functions.

In summary, we suggest that induced intramembrane cleavage of heme-binding type II transmembrane proteins leads to the release of intracellular heme sensor proteins which could regulate transcription of target genes either by direct binding to DNA or by binding to other transcription factors. SPPL2a/b would thus contribute to this signaling cascade by cleavage of homo-oligomeric TMDs, resulting in the removal of hydrophilic residues within the TMD that are important for oligomerization and therefore also for stable heme coordination. In conclusion, this study addresses a new aspect of RIP, namely the regulation of heme-dependent functions of type II transmembrane proteins in signaling and other, still to be elucidated processes. It is tempting to speculate that this mechanism also extends to type I transmembrane proteins undergoing RIP and other non-heme cofactors.

## Methods

**Expression plasmids**. Human CD74 (using the most common isoform p33 (ref. [3])), TNFα, Bri2, FasL, and Tfr1 gene sequences optimized for expression in *E. coli* were synthesized by Invitrogen GeneArt (Thermo Fisher Scientific). Cloning of full-length CD74, full-length TNFα and their N-terminal, intracellular and transmembrane domains in expression vectors was performed using standard molecular

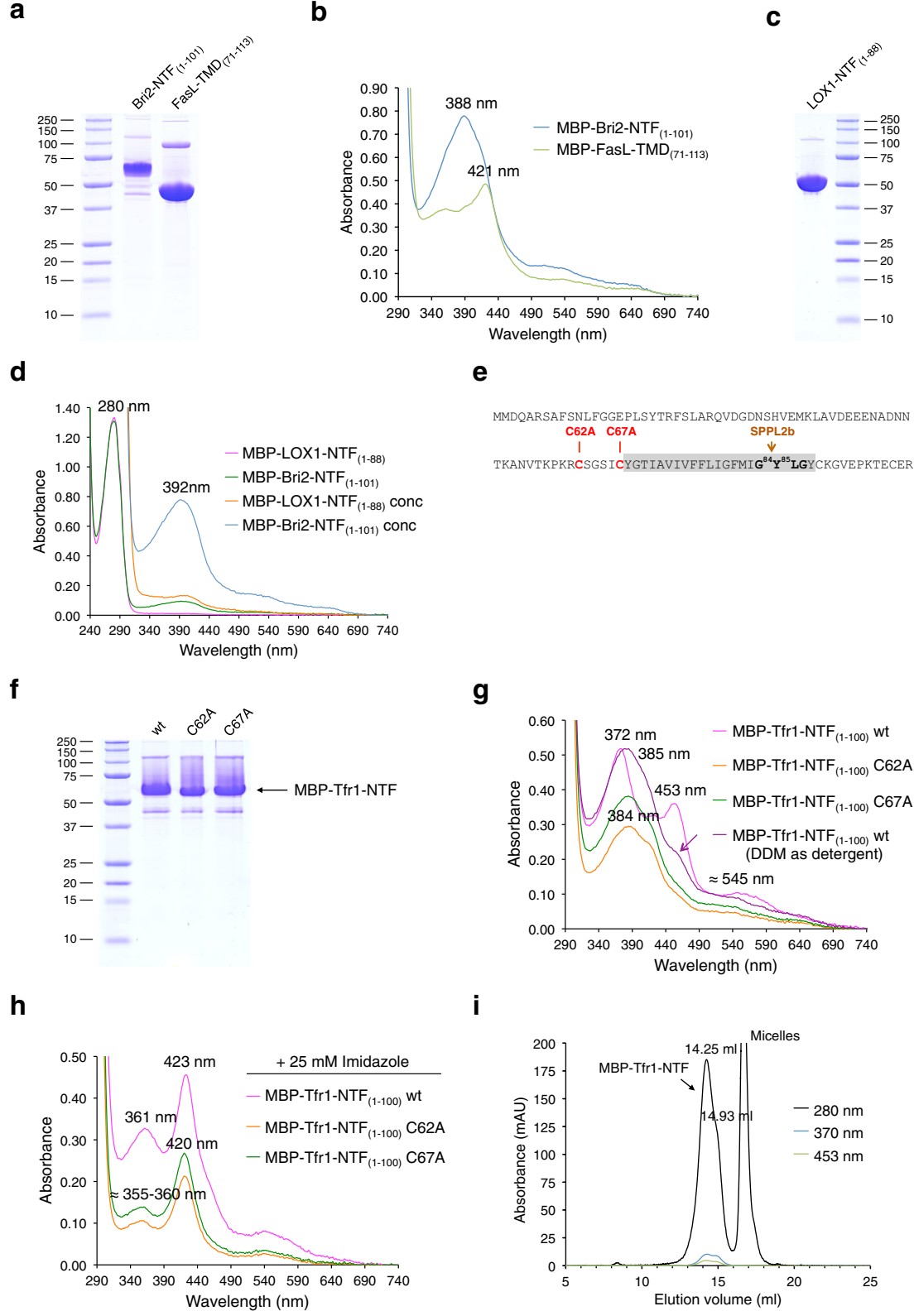

COMMUNICATIONS BIOLOGY | (2020)3:73 | https://doi.org/10.1038/s42003-020-0800-0 | www.nature.com/commsbio

biology techniques. TMD-(31–56) of CD74 was expressed together with N- and C-terminal flanking residues as 10xHis-tagged peptide [CD74-TMD$_{(25–64)}$-10xHis]. The DNA sequence of all cloned genes was confirmed using appropriate primers. DNA sequences of all constructed plasmids (listed in Supplementary Table 1) are given in Supplementary Data 1 [*NcoI/Hind*III and *NcoI/Xho*I inserts of pETDuet-1 and *SacI/Eco*RI inserts of pMalC5X (NEB)]. pMalC5X was chosen for production of type II transmembrane proteins as N-terminal MBP fusions in *E. coli*, because in this case the MBP has no signal peptide and is expressed in the cytoplasm, and

the pMalC5X encoded genetically engineered MBP exhibits tighter binding to amylose resin. Therefore, the MBP-CD74 and MBP-TNFα membrane fusion proteins have the correct topology [N-terminus in (cytoplasm) and C-terminus out (periplasm)].

**Purification of CD74-TMD$_{(25–64)}$-10xHis peptides**. We purified CD74-TMD$_{(25–64)}$ wt and C28A mutant as 10xHis fusion peptides using the LEMO

**Fig. 6 The SPPL2a/b substrates Bri2, FasL, and Tfr1 are heme-binding proteins. a** Amylose affinity purification of MBP-Bri2-NTF$_{(1-101)}$ and MBP-FasL-TMD$_{(71-113)}$ using Triton X-100 as detergent was followed by SDS-PAGE. **b** UV/VIS spectra of MBP-Bri2-NTF$_{(1-101)}$ (blue trace) and MBP-FasL-TMD$_{(71-113)}$ (green trace). Addition of imidazole shifts the Soret band of Bri2 from 388 to 418 nm (not shown). **c** Amylose affinity purification of MBP-LOX-1-NTF$_{(1-88)}$ using DDM as detergent was followed by SDS-PAGE. **d** MBP-LOX-1-NTF$_{(1-88)}$ ($\varepsilon_{280nm} = 78840\ M^{-1} cm^{-1}$) and MBP-Bri2-NTF$_{(1-101)}$ ($\varepsilon_{280nm} = 86290\ M^{-1}$ $cm^{-1}$) were purified using DDM (not absorbing at 280 nm) as detergent and analyzed before and after concentration by UV/VIS spectroscopy. **e** Sequence of Tfr1 and SPPL2b cleavage site within the GXXG motif of the TMD (in gray). Residues Cys62 and Cys67 that may contact the heme cofactor are shown in bold red. **f** Amylose affinity purification of MBP-Tfr1-NTF$_{(1-100)}$ wt and of the C62A and C67A mutants using Triton X-100 as detergent was followed by SDS-PAGE. **g** UV/VIS spectra of MBP-Tfr1-NTF$_{(1-100)}$ wt (magenta trace, split Soret spectrum with absorbance maxima at 372 and 453 nm) and the mutant proteins C62A (orange trace, Soret band at 384 nm) and C67A (green trace, Soret band at 384 nm). MBP-Tfr1-NTF$_{(1-100)}$ wt purified with DDM as detergent (violet trace) has a single Soret band at 385 nm with a shoulder at about 455 nm (indicated by violet arrow). **h** Addition of imidazole to purified MBP-Tfr1-NTF$_{(1-100)}$ proteins generates Soret bands at about 360 nm and 420–423 nm, indicating heme coordination by one Cys residue and imidazole. **i** The heme cofactor of Tfr1 is stably bound as shown by gel filtration using a Superose 6 Increase column [equilibrated in TN buffer containing 0.1% Triton X-100; absorbance is followed at 280 nm (black trace), 370 nm (blue trace), and 453 nm (green trace)]. Results in **a–d** represent one experiment. **f** and **g** show representatives out of two independent experiments, except for purification of MBP-Tfr1-NTF(1–100) using DDM which was done once. **h** Imidazole titration of wt and mutant MBP-Tfr1-NTF proteins was done once. **i** Gel filtration of wt protein to show stable heme binding was carried out once.

system for optimized membrane protein overexpression in *E. coli*[47]. For first purifications of CD74-TMD$_{(25–64)}$-10xHis wt peptide (and mutant C28A peptide, respectively), a 0.5 l culture (with 1 g glucose and 200 μM rhamnose added to the used LB medium) of *E. coli* Lemo21 (DE3) cells transformed with the appropriate plasmid was grown at 30 °C in the presence of ampicillin (100 μg ml$^{-1}$) and chloramphenicol (34 μg ml$^{-1}$). At an $A_{600}$ of about 0.5–0.6 cells were induced with 0.4 mM isopropyl 1-thio-β-D-galactopyranoside (IPTG, Sigma-Aldrich) and shaken for an additional 3 h at 30 °C. Cells were then harvested, resuspended in 15 ml TN buffer (50 mM Tris/HCl pH 7.4 at room temperature and 150 mM NaCl) containing 1 mM PMSF (Sigma-Aldrich, 100 mM stock solution in ethanol), frozen in liquid nitrogen, and stored at −80 °C. To increase heme content of the peptides[48], 0.6 mM δ-aminolevulinic acid hydrochloride (Sigma-Aldrich) was added before induction of the culture (for example, in the experiment shown in Fig. 1a–d). Thawed cells were again substituted with 1 mM PMSF and then disrupted by sonication using a Branson Sonifier 250 equipped with a microtip. The membrane pellet obtained after 30 min centrifugation at $40,000 \times g$ and 4 °C was resuspended in 15 ml TN buffer containing 1% of the detergent *N,N*-dimethyl-n-dodecylamine *N*-oxide (LDAO, Sigma-Aldrich 40236). Then, membrane proteins were solubilized at 4 °C for about 1 h in a 50 ml tube using a tube roller mixer. The supernatant obtained after centrifugation for 30 min at $40,000 \times g$ and 4 °C was subjected to immobilized metal ion affinity chromatography using Ni-NTA material (Sigma-Aldrich) equilibrated in TN buffer containing 0.1% LDAO. The column was first washed with TN/0.1% LDAO and then with TN/0.1% LDAO/20 mM imidazole. His-tagged peptides were eluted with 500 mM imidazole in TN buffer containing 0.1% LDAO and 10% glycerol and concentrated using Vivaspin 500 ultrafiltration spin columns (Sartorius). Purity of CD74-TMD peptides was increased by using higher imidazole concentrations during washing of the Ni-NTA columns. However, this led to decreased heme content of the peptides, probably because imidazole replaced the axial cysteine ligands coordinating the heme cofactor. UV/VIS spectra of the protein samples were recorded at room temperature in a 1 cm Hellma quartz absorption cuvette using the NanoDrop 2000c spectrophotometer from Thermo Fisher Scientific with elution buffer as blank.

### Purification of 10xHis-CD74-NTF$_{(1–82)}$.
10xHis-CD74-NTF-(1–82)-expressing *E. coli* Lemo21(DE3) pTK846 cells were grown at 30 °C in LB medium substituted with 200 μM rhamnose, 2 g glucose l$^{-1}$, ampicillin (100 μg ml$^{-1}$), and chloramphenicol (34 μg ml$^{-1}$). At an $A_{600}$ of about 0.6, 1.2 mM δ-aminolevulinic acid hydrochloride was added and gene expression was induced with 0.4 mM IPTG. Cells were then further shaken at 30 °C for about 3 h. Purification of 10xHis-CD74-NTF$_{(1–82)}$ was performed as described above but using Triton X-100 (for analysis, Merck 1.08603.1000; 10% (w/v) stock solution prepared in TN buffer and stored at 4 °C in the dark) instead of LDAO as membrane-solubilizing detergent and equilibrating and washing the column with 40 mM imidazole in TN buffer/0.1% Triton X-100. His-tagged peptides were eluted with 500 mM imidazole in TN/10% glycerol/0.1% Triton X-100 and concentrated using Vivaspin 500 columns.

### Purification of MBP fusion proteins from *E. coli* membranes.
MBP fusions were successfully used to increase the expression of single-spanning transmembrane proteins in *E. coli*[49]. In combination with Triton X-100 (or DDM) as detergent for membrane protein solubilization and subsequent chromatography, we also used the MBP tag for amylose affinity purification of the transmembrane proteins. *E. coli* NEB Express cells (New England Biolabs) transformed with the appropriate pMalC5X plasmid (see Supplementary Table 1) were grown in LB medium substituted with 2 g glucose l$^{-1}$ and ampicillin (100 μg ml$^{-1}$) at 30 °C. At an $A_{600}$ of about 0.5, expression was induced with 50 μM IPTG and cultures were then further shaken at 30 °C for about 3 h. Cells were harvested, resuspended in 15 ml TN buffer containing 1 mM PMSF, frozen in liquid nitrogen, and stored at −80 °C. For the experiments shown in Figs. 2d–g, 3b–g, 5, 6a–d and f–i and Supplementary Figs. 4,

5f–i, 6, and 7, 0.6–1.2 mM δ-aminolevulinic acid hydrochloride was added before induction with IPTG. Thawed cells were again substituted with 1 mM PMSF and then disrupted by sonication. The membrane pellet obtained after 30 min centrifugation at $40,000 \times g$ and 4 °C was resuspended in 15 ml TN buffer containing 1% of the detergent Triton X-100. Then, membrane proteins were solubilized at 4 °C for about 1 h in a 50 ml tube using a tube roller mixer. The supernatant obtained after centrifugation for 30 min at $40,000 \times g$ and 4 °C was subjected to amylose (New England Biolabs) affinity chromatography at room temperature. Amylose affinity columns were equilibrated and washed with TN buffer containing 0.1% Triton X-100. MBP fusion proteins were eluted with 10 mM maltose (Sigma-Aldrich) in equilibration buffer, concentrated using Vivaspin 500 columns, and snap frozen in liquid nitrogen. If DDM (Avanti) was used instead of Triton X-100, solubilization of membrane proteins was carried out with 2% detergent and amylose affinity chromatography was done with 0.2% detergent in TN buffer. UV/VIS spectra of (if not otherwise indicated) concentrated MBP fusion proteins were recorded as described above. To reduce ferric heme to ferrous heme, a few microliters of a concentrated sodium dithionite (Riedel de Haën) solution in TN buffer were added to the heme protein.

### Hemin reconstitution before affinity chromatography.
In contrast to pathogenic *E. coli* strains, the used laboratory K12 strains are not able to degrade heme. This enables heme reconstitution of MBP fusion proteins before amylose affinity chromatography. To cytoplasmic protein solutions in TN buffer containing either MBP-CD74-ICD$_{(1–42)}$ wt, MBP-CD74-ICD$_{(1–42)}$ C28A, or MBP-TNFα-ICD$_{(1–34)}$ fusion proteins 50 μM hemin (=ferric chloride heme, Sigma-Aldrich; 5 mM stock solution in DMSO) was added. After incubating the hemin-containing protein solution for about 20 min at room temperature in a 50 ml tube using a roller mixer, amylose chromatography was carried out. MBP fusion proteins eluted in TN buffer containing 10 mM maltose were analyzed by UV/VIS spectroscopy and SDS-PAGE.

### Hemin reconstitution and dissociation of TNFα-(1–39) peptide.
TNFα-(1–39) peptide H$_2$N-MSTESMIRDVELAEEALPKKTGGPQGSRRCLFLSLFSFL-COOH was synthesized by Thermo Fisher Scientific with a purity of 95% and delivered as lyophilized, trifluoroacetic salt. Solubility of the peptide in TN buffer was low, e.g. it was not possible to solve 50 μM peptide in TN buffer. Already, the experiments with MBP-TNFα-ICD$_{(1–39)}$ showed high tendency of the peptide to form cluster in the absence of heme (even in presence of detergent). We assume that bis-thiolate ligation of heme by TNFα-(1–39) leads to refolding of the peptide and formation of a structured heme-containing dimer. Therefore, we added 1 ml 25 μM hemin in TN buffer (for some experiments containing 0.1% Triton X-100) directly to about 1 mg of the lyophilized peptide and mixed either at room temperature using a Vortex mixer or at 37 °C using an Eppendorf ThermoMixer. After centrifugation for 5 min at $20,800 \times g$ UV/VIS spectra of the supernatants were recorded; the light brown pellet was discarded. For dissociation experiments, lyophilized synthetic TNFα-(1–39) peptide was incubated with hemin in TN buffer adjusted to pH 8.8 (to increase solubility of hemin) at room temperature, centrifuged for 5 min at $20,800 \times g$, and then horse apomyoglobin (Sigma-Aldrich A8673) was added to the supernatant containing the heme-coordinated peptide using a molar ratio of apomyoglobin:hemin of about 2:1 [due to the experimental approach we cannot exclude that peptide not ligated with heme is already present before addition of apomyoglobin; however, because of high-affinity heme binding by apomyoglobin ($K_D$ of about $3 \times 10^{-15}$ M)[50] this will not change the conclusion drawn from the dissociation experiment].

### Gel filtration.
Gel filtration was carried out at 4 °C using an ÄKTA pure system from GE Healthcare and a Superose 6 Increase 10/300 GL column for high-

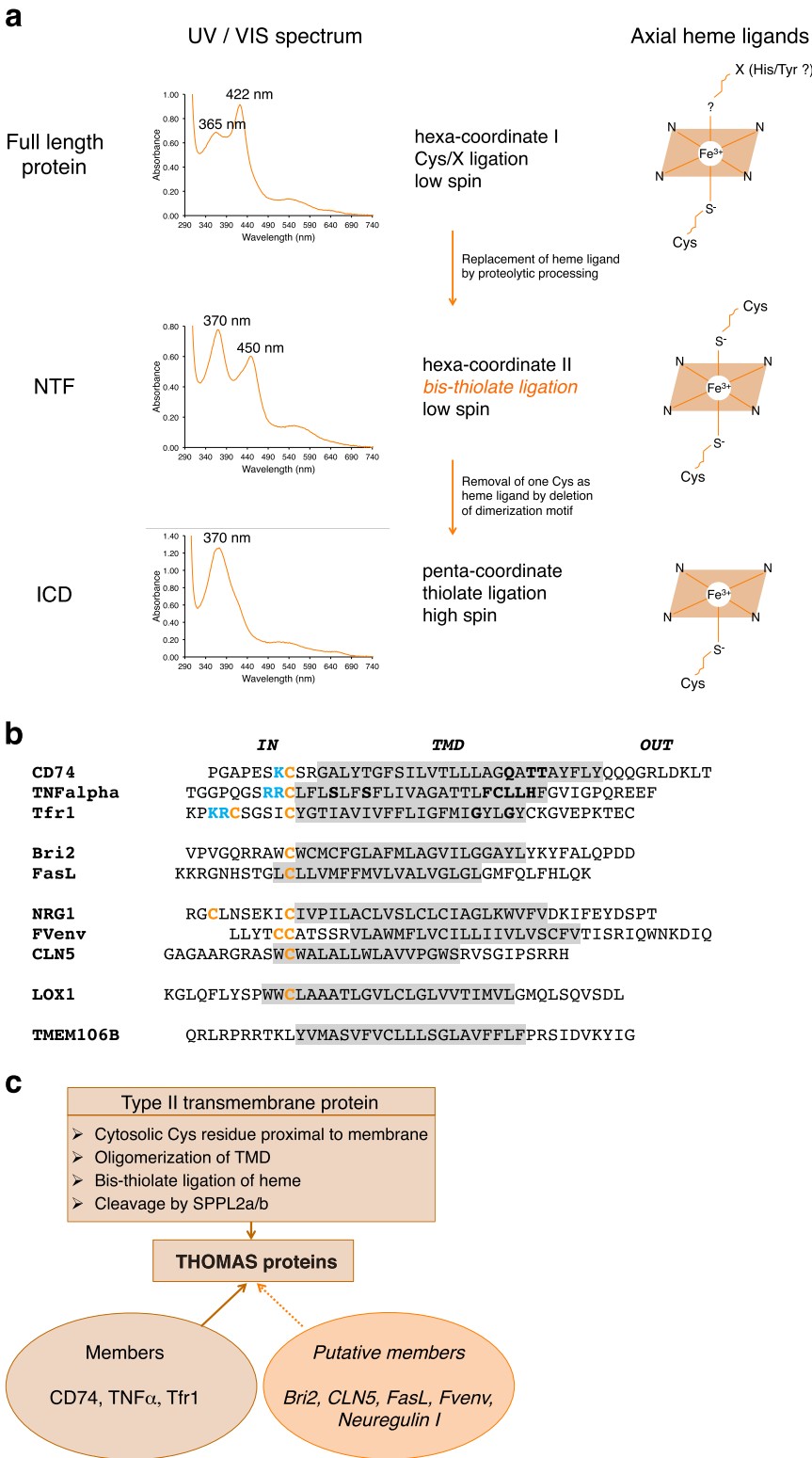

**Fig. 7 The THOMAS protein family. a** Scheme of different modes of heme coordination by proteolytically processed type II transmembrane proteins. **b** Sequence comparison of TMDs (shadowed in gray; including NH₂- and COOH-terminal flanking residues) of type II transmembrane proteins cleaved by SPPL2a/b. In the present study we show that the SPPL2a/b substrates CD74, TNFα, Tfr1, Bri2, and FasL are heme-binding proteins. Cytosolic cysteine residues (in orange; preceded by basic residues highlighted in blue) are axial heme ligands of CD74, TNFα, and Tfr1. Residues of the TMD which are known to be important for oligomerization are shown in bold black letters. The TNFα TMD contains an SXXS dimerization motif and in addition an FCLLH motif, which we suggest is important for trimerization (compare Supplementary Fig. 7). Cleavage by SPPL2a/b removes these oligomerization motifs. Length of TMD varies depending on the prediction tool. **c** The THOMAS (**T**hiolate-**H**eme **O**ligomeric type II trans-**M**embrane proteins with heme binding mode **A**djusted by **S**PPL2a/b) protein family. N-terminal fragments of CD74, TNFα, and Tfr1 are characterized by bis-thiolate ligation of heme. It has to be investigated if proteolytically processed putative members of the THOMAS family like Bri2, CLN5, FasL, Fvenv, and Neuregulin I also bind heme by bis-thiolate ligation. LOX-1 is binding heme to a very low extent and therefore is not included into the THOMAS family.

**a**

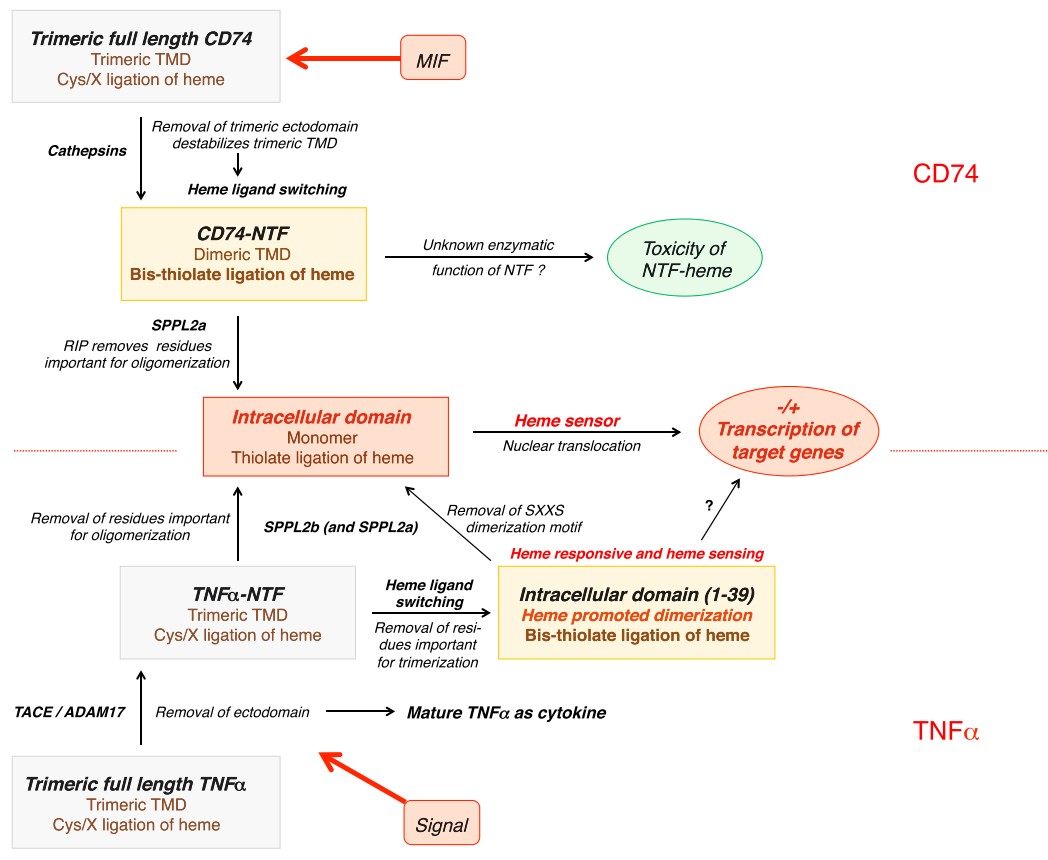

**b**

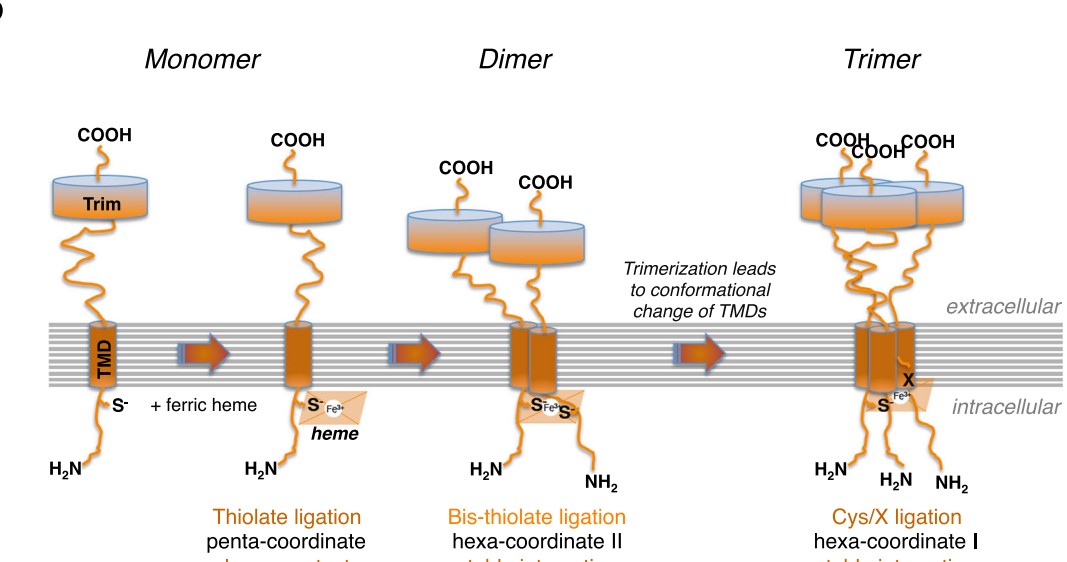

**Fig. 8 Regulated intramembrane proteolysis and biogenesis of heme-binding proteins of the THOMAS family. a** Regulated intramembrane proteolysis changes the heme binding mode of type II transmembrane proteins CD74 (upper part of the figure) and TNFα (lower part) and generates putative intracellular heme sensor domains. **b** Model for the biogenesis of trimeric THOMAS proteins. Assembly starts with a monomeric membrane spanning protein (Trim lumenal trimerization domain, TMD transmembrane domain) that loosely contacts ferric heme with one cysteine thiolate as the only axial ligand (penta-coordinate ferric heme). Dimerization of the monomeric transmembrane protein leads to stable bis-thiolate ligation (hexa-coordinate ferric heme). Interaction with a third monomer then leads to trimerization both of the soluble luminal domains and of the transmembrane domains, triggering a conformational change within the TMDs. This conformational change results in a heme ligand switching from two heme-binding cysteine residues to only one cysteine and in addition a so far unidentified residue (X).

resolution size exclusion chromatography. MBP fusion proteins (for most experiments concentrated with Vivaspin 500 columns) were slowly thawed on ice and centrifuged for 5 min (20,800 × $g$) at 4 °C and then 100 μl of protein aliquots of the supernatant were subjected to the column equilibrated in TN buffer containing 0.1% Triton X-100 at a flow rate of 0.4 ml min$^{-1}$. The elution was followed by absorbance at 280 nm and simultaneously for the detection of heme at additional wavelengths (such as 370, 422, and 450 nm), and 0.4 ml fractions were collected. Gel filtration of (hemin-reconstituted) soluble MBP-CD74-ICD$_{(1-42)}$ and MBP-TNFα-ICD$_{(1-34)}$ proteins was carried out with TN buffer containing no detergent. The void volume of the used column was determined with Blue dextran 2000 (GE Healthcare) to be 8.5 ml.

**SDS-PAGE and immunoblotting.** Proteins were separated using Tricine-SDS-polyacrylamide (10%) gel electrophoresis under reducing (1× sample buffer containing 10 mM DTT) conditions[51]; an all blue prestained protein standard was obtained from Bio-Rad. Molecular weights of standard proteins are marked on the side of the gel pictures. For documentation, destained gels were scanned using an Epson Perfection V850 Pro scanner with 1200 d.p.i. resolution. Proteins were electrophoretically transferred to PVDF membrane (Immobilon-FL, Merck Millipore) after SDS-PAGE by semi-dry blotting. MBP fusions were detected with a monoclonal anti-MBP antibody (New England Biolabs, diluted 1:4000) as first antibody, an Alexa 680 goat-anti-mouse antibody (Life Technologies, Thermo Fischer Scientific, 0.8 μg ml$^{-1}$) as secondary antibody and then using the Odyssey Imaging System (Li-Cor Biosciences).

**Electron paramagnetic resonance.** MBP-TNFα-ICD$_{(1-39)}$ L31P was purified from 1 l IPTG (50 μM) induced *E. coli* NEB Express pMalC5X-*TNFα-ICD-(1–39)-L31P* cells grown overnight at 20 °C in the presence of 1.2 mM δ-aminolevulinic acid hydrochloride as described above with the following changes. For solubilization of the membrane fraction 2% DDM in TN buffer was used. After applying the protein solution the amylose column was first washed with 0.2% DDM and then with 0.02% DDM in TN buffer. MBP-TNFα-ICD$_{(1-39)}$ L31P was eluted with 10 mM maltose in TN buffer containing 0.02% DDM, concentrated about 10-fold using Vivaspin 500 columns, and then adjusted to 15% glycerol. Using the published molar absorption coefficient $\varepsilon_{450}$ of about 70,000 M$^{-1}$ cm$^{-1}$ for bis-thiolate-ligated ferric heme of DGCR8 (ref. [17]) heme content of this protein solution was determined to be about 130 μM. About 50 μl of this protein solution was filled into Suprasil EPR quartz capillaries with 3 mm inner diameter and frozen using liquid nitrogen. Continuous wave EPR measurements were then performed in Osnabrück on a Bruker Elexsys E580 EPR spectrometer operating at X-band (~9.4 GHz), equipped with a Bruker SHQ Super High Sensitivity Probehead (Bruker Biospin; Germany). EPR spectra were recorded at 10 K. For stabilization of the sample temperature a continuous flow cryostat Oxford ESR900 (Oxford Instruments, Oxfordshire, UK) and a temperature controller (ITC 3; Oxford Instruments) was used. The EPR spectra were obtained with a sweep time of 83 s with a time constant of 20.48 ms, a field modulation amplitude of 5 G, a modulation frequency of 100 kHz, and 1 mW microwave power. The scan range was 1300 Gauss, centered at 3050 Gauss.

**Statistics and reproducibility.** No statistical analyses were performed. Regarding reproducibility it is important to use Triton X-100 solutions that are freshly prepared and stored in the dark for protein purifications, since polyoxyether detergents otherwise may contain high peroxide concentrations changing the UV/Vis spectra of heme proteins. wt and mutant proteins were always purified in parallel using the same purification procedure including the same buffers and detergent solutions (and using cells grown under the same conditions). The precise numbers of purifications are indicated in the figure legends.

**Reporting summary.** Further information on research design is available in the Nature Research Reporting Summary linked to this article.

## Data availability

All relevant data are available in the paper, Supplementary Information files (sequences of cloned DNA fragments can be found in Supplementary Data 1), and from the corresponding authors upon request.

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

## Acknowledgements

The project was funded by the Deutsche Forschungsgemeinschaft (DFG, German Research Foundation)—Project Number 319506281—TRR 186, and Project Number 283963326. B.B. is member of the DFG Excellence Cluster CellNetworks Exc81. We thank Alexia Herrmann for expert technical assistance, Heinz-Jürgen Steinhoff for helpful discussions, and Penelope Kay-Jackson, Rainer Beck, and Walter Nickel for commenting on the manuscript.

## Author contributions

T.K. discovered heme binding of CD74, TNFα, Bri2, Tfr1, and FasL and designed the overall research in coordination with B.B. T.K. performed all experiments apart from the EPR spectroscopy experiment shown in Fig. 5h, which was performed by J.K. T.K. prepared the figures, and T.K. and B.B. wrote the manuscript. All authors commented on the manuscript.

## Competing interests

The authors declare no competing interests.
