## [Peer Review File · Communications Biology]

Reviewers' comments:

Reviewer #1 (Remarks to the Author):

Overall, the manuscript from Kupke and Brügger is thought-provoking, well performed, rigorous research. In this study, the researchers found that several SPPL2 substrates co-ordinate heme. They demonstrate that the requirements for this event are oligomerisation of the substrate transmembrane segments and a cytoplasmic, membrane-proximal cysteine residue. Most interestingly, they found that different fragments (produced in vitro), which are produced in vivo by processive/sequential trimming of the substrate(s) alters the mode of heme binding. These are very original, unexpected insights and heme binding to these fragments may have important biological roles in vivo, once released from the membrane. For instance, they may translocate to the nucleus to modulate gene expression or play enzymatic roles in the cytoplasm. Given the importance of heme in human biology, this opens multiple new avenues of investigation. In general, the research is well presented, and the arguments and statements are supported by the data. Likewise, when the authors speculate, they mostly clearly state so.

The only (minor) concern I have is that the title and the abstract do not indicate clearly that this study is wholly in vitro, and therefore they are unwittingly slightly misleading. This is an in vitro research study, where the fragments that are generated by proteolysis in vivo are synthesised in vitro, thus not generated by the proteases themselves. There is no evidence that the process of regulated intramembrane proteolysis do in fact generate fragments that bind heme ; what this study shows is that peptides that mimic the products of regulated intramembrane proteolysis have different modes of heme co-ordination. It clearly shows that what the authors propose is possible – but whether this event occurs in vivo, and which cells have levels of heme to support such an event, are unclear (I do note that the peptides display high affinity binding to heme, and that cells have levels of heme in the nanomolar range). This may sound like a small distinction, but in reality, I feel that the title and claims in the abstract could be toned down and be more specific. For instance, in the abstract, it is stated that "Proteolytic processing of CD74 and TNFalpha changes the mode of heme binding". Such a statement is not supported – and in a couple of places in the manuscript, this is the case. Essentially, it is the difference between what something can do in vitro, versus what it does do in vivo. At the end, in Figure 6, it is clear that this is a speculative model that is being proposed – which is fine, but the sequence of events, although consistent with the data, are an interpretation stretch, given the data. Last, the title does not even mention heme as the cofactor, or that this is specific to SPPL2 substrates. Perhaps the authors think that the more generalised title/abstract is more attractive, but I found it a bit disappointing/frustrating at first. Once I read the paper fully, I felt that I experienced great science that spoke for itself.

Overall, I think the paper should be accepted if those changes were made to the text, I genuinely enjoyed reading it and think it is important science. I believe these results are important for the field of membrane protein proteolysis, and due to the variety of roles played by the substrates, and the potential regulatory roles for heme coordinated peptides, it is also of interest to a broader readership. Therefore I do not have any essential additional experiments. What I do have are some thoughts and suggestions of experiments, which I leave down to the authors' judgement to decide whether they are worth doing to improve the manuscript.

Suggestions/Comments/Questions:

1. Somewhere, a chemical scheme of the different modes of heme co-ordination would aid understanding of the paper.

2. Remove Fig 1A – as it confuses the message a little (the C28A peptide appears almost as brown at the WT). I initially thought that this meant that it still coordinated heme. The data that follow are super clear. For example, in comparison, the C30A mutant in Fig 2D is completely clear, compared to the brown WT.
3. Would any transmembrane domain with a dimerization motif (GxxxG) and a membrane proximal cysteine co-ordinate heme? If so, it could widen the scope of the findings in this manuscript – and support the claim that substrates of other intramembrane proteases may also co-ordinate heme.
4. In the experiments presented on the dimerization of MBP-TNFalpha-ICD(1-39) – in Figure 3g and Suppl. Fig 6 – there seems to be less coordinate heme, and less dimer. Did the authors consider a double serine mutant? Perhaps this would abolish all dimerization/heme binding?
5. Are gels (as in Fig 3d) available of apomyoglobin treated MBP-TNFalpha-ICD(1-39) (as in Fig 4f/g)? This would presumably clearly show the loss of dimerization upon loss of heme coordination.
6. Can a legend be placed in Fig 4f/g? This would make the figure more understandable, stand-alone.
7. Why are there single sort bands for Bri2 and FasL, unlike Tfr1? Can the authors make a comment on, or discuss this?
8. Have the authors mutated the cysteines in Bri2, FasL and Tfr1 to demonstrate their role in heme co-ordination?
9. Could the authors include an alignment of the TMDs+regulatory cysteines of THOMAS proteins? Are there any conserved elements? Would this highlight a conserved residue (which is as yet unidentified) that is responsible for the heme ligand switching event at the trimerization step proposed in Fig 6A?
10. Did the authors ever test MBP-CD74(1-216)-C28A?
11. For Figure 2F, is it possible to produce the graphs as overlay comparisons with WT? This would allow the reader to clearly see the UV/VIS deviations.
12. Is it possible the purify SPPL2-generated fragments from mammalian cells and show that they co-ordinate heme? I appreciate this may be a next step for the lab.

Reviewer #2 (Remarks to the Author):

The study of Kupke & Brugger provides evidence that two related members of a family of transmembrane cytosolic-cleavable proteins (CD74 & TNFa) can unexpectedly act as heme-binding proteins, and that their cleaved intracellular fragments can also be heme-binding. They show that the cleavage itself, and point of cleavage, as well as specific Cys, are all important for determining the heme-binding mode of the fragments, and in some cases for determining their oligomer status. The authors speculate that heme binding as seen in their E-coli-based or purified protein systems may be possible in mammalian cells, and may have an unappreciated but important impact on the signal transduction cascades that the fragments are involved in. Overall, their research findings seem accurate and are unexpected, quite provocative, and might be a fundamental advance in the field.

Comments/Concerns:

1. The enthusiasm is dampened somewhat by not showing evidence for fragment heme-binding occurring in a mammalian cell. Is this not feasible? It would greatly improve the impact of the study.
2. Assigning heme axial ligands by UV-vis data alone is not as clear-cut as the authors seem to imply. Could some other spectroscopic methods (R Raman, EPR) be employed to confirm the axial ligand assignments for at least the main fragments of interest?

Reviewer #1

Overall, the manuscript from Kupke and Brügger is thought-provoking, well performed, rigorous research. In this study, the researchers found that several SPPL2 substrates co-ordinate heme. They demonstrate that the requirements for this event are oligomerisation of the substrate transmembrane segments and a cytoplasmic, membrane-proximal cysteine residue. Most interestingly, they found that different fragments (produced in vitro), which are produced in vivo by processive/sequential trimming of the substrate(s) alters the mode of heme binding. These are very original, unexpected insights and heme binding to these fragments may have important biological roles in vivo, once released from the membrane. For instance, they may translocate to the nucleus to modulate gene expression or play enzymatic roles in the cytoplasm. Given the importance of heme in human biology, this opens multiple new avenues of investigation. In general, the research is well presented, and the arguments and statements are supported by the data. Likewise, when the authors speculate, they mostly clearly state so.

The only (minor) concern I have is that the title and the abstract do not indicate clearly that this study is wholly in vitro, and therefore they are unwittingly slightly misleading.

This is an in vitro research study, where the fragments that are generated by proteolysis in vivo are synthesised in vitro, thus not generated by the proteases themselves. There is no evidence that the process of regulated intramembrane proteolysis do in fact generate fragments that bind heme ; what this study shows is that peptides that mimic the products of regulated intramembrane proteolysis have different modes of heme co-ordination. It clearly shows that what the authors propose is possible – but whether this event occurs in vivo, and which cells have levels of heme to support such an event, are unclear (I do note that the peptides display high affinity binding to heme, and that cells have levels of heme in the nanomolar range). This may sound like a small distinction, but in reality, I feel that the title and claims in the abstract could be toned down and be more specific. For instance, in the abstract, it is stated that “Proteolytic processing of CD74 and TNFalpha changes the mode of heme binding”. Such a statement is not supported – and in a couple of places in the manuscript, this is the case. Essentially, it is the difference between what something can do in vitro, versus what it does do in vivo. At the end, in Figure 6, it is clear that this is a speculative model that is being proposed – which is fine, but the sequence of events, although consistent with the data, are an interpretation stretch, given the data.

Last, the title does not even mention heme as the cofactor, or that this is specific to SPPL2 substrates. Perhaps the authors think that the more generalised title/abstract is more attractive, but I found it a bit disappointing/frustrating at first. Once I read the paper fully, I felt that I experienced great science that spoke for itself.

We thank the reviewer for the constructive criticism. In order to avoid any misapprehension, we have changed the title to:

‘Heme binding of transmembrane signaling proteins undergoing regulated intramembrane proteolysis‘.

We also now clearly state in the abstract that this is an in vitro study by adding the following information on page 2:

To study the role of transmembrane domains (TMDs) in modulating structure and activity of the fragments released by RIP we recombinantly expressed the type II transmembrane proteins CD74 and TNFalpha and their processing products in *Escherichia coli*.

Overall, I think the paper should be accepted if those *changes were made to the text*, I genuinely enjoyed reading it and think it is important science. I believe these results are important for the field of membrane protein proteolysis, and due to the variety of roles played by the substrates, and the potential regulatory roles for heme coordinated peptides, it is also of interest to a broader readership. *Therefore I do not have any essential additional experiments*. What I do have are some thoughts and suggestions of experiments, which I leave down to the authors' judgement to decide whether they are worth doing to improve the manuscript.

Suggestions/Comments/Questions:

1. Somewhere, a chemical scheme of the different modes of heme co-ordination would aid understanding of the paper.

We included this scheme as new Figure 7a.

2. Remove Fig 1A – as it confuses the message a little (the C28A peptide appears almost as brown at the WT). I initially thought that this meant that it still coordinated heme. The data that follow are super clear. For example, in comparison, the C30A mutant in Fig 2D is completely clear, compared to the brown WT.

The C28A peptide still binds very low amounts of heme. We changed Figure 1a, and now the different amount of heme bound by the two peptides as also observed in Figure 1 d is good to see.

3. Would any transmembrane domain with a dimerization motif (GxxxG) and a membrane proximal cysteine co-ordinate heme? If so, it could widen the scope of the findings in this manuscript – and support the claim that substrates of other intramembrane proteases may also co-ordinate heme.

The analysis of additional substrates of intramembrane proteases is an interesting topic, and we have not yet looked at a sufficiently large number of proteins that would allow a more general statements on heme-binding properties. One of the proteins we have included into this study is LOX-1, a type II transmembrane protein of the family of C-type lectin receptors. Like CD74 and TNFalpha, LOX-1 is a substrate of SPPL proteases. It contains a GxxxG-like motif (AxxxG, and a leucine in position -1, see Senes et al., *JMBiol* 2000; 296:921-936). In this case only very low heme amounts were copurified. However, in LOX-1 the cysteine residue is within the N-terminal region of the TMD. We included this experiment as new Figure 6 c and d. Other substrates of SPPL2a/b and also type I transmembrane proteins are subject of ongoing studies.

4. In the experiments presented on the dimerization of MBP-TNFalpha-ICD(1-39) – in Figure 3g and Suppl. Fig 6 – there seems to be less coordinate heme, and less dimer.

It is correct that the absorbance at 451 nm of the mutants ICD-(1-39) S34A and ICD-(1-39) S37A is decreased compared to ICD-(1-39) wt, showing that less bis-thiolate ligated heme is present. This can be explained by the lower amounts of dimers observed (see Figure S6c-g) that go along with a lower amount of total heme bound. We conclude that dimerization is important for stable heme binding. However, mutation of the membrane proximal cysteine residue (C30A) that leads to a loss of heme binding also affects dimerization of MBP-TNFalpha-ICD-(1-39) (see Figure 3d-e, and Figure S5f). Therefore, as had already been included into our manuscript heme promoted dimerization involves the SXXS motif as well as the axial heme ligand Cys30.

Did the authors consider a double serine mutant? Perhaps this would abolish all dimerization/heme binding?

We agree with the reviewer that loss of both serine mutants should completely abolish heme promoted dimerization of TNFalpha-ICD. Instead of analyzing this double mutant we have analyzed ICD-(1-34) in which Ser37 and adjacent residues are deleted, and which, like ICD-(1-39) is generated *in vivo* by intramembrane proteolysis. This monomeric ICD is purified without heme, but can be reconstituted with heme (Figure 3h-j), confirming that the motif S³⁴LFS³⁷FL is important for bis-thiolate ligation of heme.

5. Are gels (as in Fig 3d) available of apomyoglobin treated MBP-TNFalpha-ICD(1-39) (as in Fig 4f/g)? This would presumably clearly show the loss of dimerization upon loss of heme coordination.

Since the heme cofactor is not covalently bound to MBP-TNFalpha-ICD-(1-39), heme and protein are separated during denaturing conditions of SDS-PAGE. It is not clear why we see a weak dimer signal for MBP-TNFalpha-ICD-(1-39) wt in Figure 3d. One explanation could be that due to the presence of heme (and therefore iron) the Fenton reaction takes place and formed radicals lead to covalent cross-linking of protein monomers. This possibility is discussed in the legend to Supplementary Figure 3b.

6. Can a legend be placed in Fig 4f/g? This would make the figure more understandable, stand-alone.

We have included the legend to Fig. 4f. In addition, we included a control experiment as new Figure 4g. The original Figure 4g was edited and is now Figure 4h. For quantification we now only used the more reliable absorbance increase at 409 nm and the data were then fitted with the GraphPad Prism program using the *exponential one phase decay function*.

7. Why are there single soret bands for Bri2 and FasL, unlike Tfr1? Can the authors make a comment on, or discuss this?

The heme ligation of both proteins, indeed, will have to be investigated in more detail in future experiments. In the present manuscript we only describe their heme binding property. It is possible that FasL-TMD forms trimers or higher oligomers and not dimers as described for Tfr1, and this might be the reason that we do not see bis-thiolate ligation for FasL.

8. Have the authors mutated the cysteines in Bri2, FasL and Tfr1 to demonstrate their role in heme co-ordination?

We mutated the cysteine residues Cys62 and Cys67 of Tfr1 and show that both residues have a role in heme coordination (new Figure 6 e-i).

9. Could the authors include an alignment of the TMDs+regulatory cysteines of THOMAS proteins?

This alignment was introduced as new Figure 7b.

Are there any conserved elements?

There are no conserved elements within the amino acid sequence, but in case of CD74, TNFalpha and Tfr1 heme binding cysteine residues are preceded by basic residues (see new Figure 7b).

Would this highlight a conserved residue (which is as yet unidentified) that is responsible for the heme ligand switching event at the trimerization step proposed in Fig 6A

No, this is still an open question. However, we think (supported by published data) that there are different oligomerization motifs within the TMDs of the investigated type II transmembrane proteins. In TNFalpha, for example, we identified the SXXS motif as dimerization motif and the FCLLH motif as being important for trimerization (see new Figure 7b). To identify conserved residues that are responsible for the heme ligand switching event is an important task which we are interested to address in future experiments.

10. Did the authors ever test MBP-CD74(1-216)-C28A?

A MBP fusion protein of full length CD74-(1-216) C28A was cloned and tested for recombinant expression in *Escherichia coli*, but we have not been successful in establishing conditions with the protein not being subjected to degradation so far.

11. For Figure 2F, is it possible to produce the graphs as overlay comparisons with WT? This would allow the reader to clearly see the UV/VIS deviations.

We changed Figure 2f as suggested and introduced the UV/VIS spectrum of H52A mutant in presence of imidazole as Figure 2g.

12. Is it possible the purify SPPL2-generated fragments from mammalian cells and show that they co-ordinate heme? I appreciate this may be a next step for the lab.

So far we have not been successful in purifying Twin-Strep-Tag II and Flag tagged NTFs of TNFalpha and CD74 and Twin-Strep-Tag II tagged TNFalpha-ICD-(1-39) (wt/C30A/L31P) in amounts suitable for biochemical characterizations due to their extremely low concentration in mammalian cells. Because of the transient heme interaction soluble heme sensor peptides like CD74-ICD-(1-42) and TNFalpha-ICD-(1-34) can only be reconstituted with heme – as has already been shown in our paper.

Reviewer #2:

The study of Kupke & Brugger provides evidence that two related members of a family of transmembrane cytosolic-cleavable proteins (CD74 & TNF α) can unexpectedly act as heme-binding proteins, and that their cleaved intracellular fragments can also be heme-binding. They show that the cleavage itself, and point of cleavage, as well as specific Cys, are all important for determining the heme-binding mode of the fragments, and in some cases for determining their oligomer status. The authors speculate that heme binding as seen in their E-coli-based or purified protein systems may be possible in mammalian cells, and may have an unappreciated but important impact on the signal transduction cascades that the fragments are involved in. Overall, their research findings seem accurate and are unexpected, quite provocative, and might be a fundamental advance in the field.

Comments/Concerns:

1. The enthusiasm is dampened somewhat by not showing evidence for fragment heme-binding occurring in a mammalian cell. Is this not feasible? It would greatly improve the impact of the study.

We agree with the reviewer that purification of heme-binding fragments from mammalian cells is an important task. However, purification of signaling peptides and peptides including transmembrane domains is also known to be a very challenging task. We expressed different RIP processing products in mammalian cells but have not yet been successful in purifying tagged NTFs of TNF α and CD74 and tagged TNF α -ICD-(1-39) (wt/C30A/L31P) peptides in amounts suitable for biochemical characterizations due to their extremely low concentration in mammalian cells. Because of the transient heme interaction soluble heme sensor peptides like CD74-ICD-(1-42) and TNF α -ICD-(1-34) can only be reconstituted with heme – as has been shown in our manuscript.

2. Assigning heme axial ligands by UV-vis data alone is not as clear-cut as the authors seem to imply.

In our manuscript we have used UV/VIS spectroscopy to identify heme axial ligands but also site-directed mutagenesis to show that all investigated fragments are so-called heme-thiolate proteins (Smith, A.T. et al. *Chem Rev* **115**, 2532-58 (2015)) with at least one Cys residue as axial heme ligand.

So far DGCR8 displaying a unique split Soret spectrum with maxima at about 370 and 450 nm is the only protein for which a bis-thiolate ligation of heme was shown: in a dimer each monomer contributes one cysteine residue for heme coordination. However, there are other examples for heme bis-thiolate ligation with split Soret spectra similar to that of DGCR8 as described in

- a) Ruf, H.H. & Wende, P. Hyperporphyrin spectra of ferric dimercaptide-hemin complexes. Models for ferric cytochrome P450-thiol complexes. *Journal of the American Chemical Society* **99**, 5499-5500 (1977).
- b) Sono, M., Dawson, J.H. & Hager, L.P. The generation of a hyperporphyrin spectrum upon thiol binding to ferric chloroperoxidase. Further evidence of endogenous thiolate ligation to the ferric enzyme. *J Biol Chem* **259**, 13209-16 (1984).

- c) Frydenvang, K. et al. Structural analysis of Cytochrome P450 BM3 mutant M11 in complex with dithiothreitol. *PLoS One* **14**, e0217292 (2019).

These publications support our view that CD74-NTF and TNFalpha-ICD-(1-39) are binding heme via bis-thiolate ligation. In the revised version of our manuscript we have included these references.

However, to further support bis-thiolate ligation of heme we also included additional experiments to the revised version of the manuscript:

- a) For structural studies like EPR we aimed to increase the heme content of TNFalpha-ICD-(1-39). It has been shown that a Pro residue directly following a cysteine residue is increasing the binding affinity of the Cys residue for ferric heme. Therefore, we generated an additional construct with Cys30Leu31 exchanged for Cys30Pro31. As predicted we observed an increased heme content and stability of the bis-thiolate heme complex (see new Figure 5) purified from *Escherichia coli*, supporting that Cys30 is an axial ligand of heme. We used this mutant to assign the axial ligands by EPR (see below).
- b) Within Tfr1-NTF we exchanged both cytoplasmic Cys residues (Cys62 and Cys67) for Ala. Interestingly both single mutations disrupt the split Soret spectrum and thus bis-thiolate ligation, indicating that the axial ligands of heme are in this case Cys62 and Cys67. This is also supported by the fact that removal of only one Cys residue does not abolish heme binding of Tfr1-NTF.

Could some other spectroscopic methods (R Raman, EPR) be employed to confirm the axial ligand assignments for at least the main fragments of interest?

As suggested by the reviewer we performed EPR studies to allow ligand assignment. Both wt and TNFalpha-ICD-(1-39) L31P showed similar UV/Vis spectra with absorbance maxima at 371 and 451 nm. Since high heme (= spin) concentrations are necessary for EPR experiments we used the L31P mutant for further analyses. We observed *g* values of 2.40, 2.28 and 1.91 (new Figure 5g; and in a second independent experiment of 2.41, 2.28 and 1.92). The EPR data confirm low-spin thiol coordination and further support bis-thiolate ligation of heme (compare EPR properties of P-450-CAM sulfur donor complexes as published in Dawson *et al.*, *J. Biol. Chem.* 1982, 3606-3617 and reviewed in Dawson and Sono in *Chem. Rev.* 1987, 1255-1276).

REVIEWERS' COMMENTS:

Reviewer #1 (Remarks to the Author):

Overall, Kupke, Klare and Brügge have done a fantastic job with this paper and I am fully satisfied with their amendments. The original finding, that various substrates of SPPL2 can co-ordinate heme, will be of significant interest to the intramembrane protease field, and to those that investigate associated cell signaling processes. The authors have gone above and beyond what was required – and the changes, especially to the figures, have improved the clarity of the paper. I look forward to seeing this paper in press – and thanks to the authors for their careful consideration of my comments.

Reviewer #2 (Remarks to the Author):

The authors did a good job in addressing my questions and concerns.